# AN INFORMATION THEORETIC PERSPECTIVE ON AGENTIC SYSTEM DESIGN

**Shizhe He**[1], **Avanika Narayan**[1], **Ishan S. Khare**[1], **Scott W. Linderman**[2,3], **Christopher Ré**[1]
**Dan Biderman**[1,2,3]
[1]Department of Computer Science, Stanford University
[2]Department of Statistics, Stanford University
[3]Wu Tsai Neurosciences Institute, Stanford University
`shizhehe@stanford.edu`

## ABSTRACT

Agentic language model (LM) systems power modern applications like "Deep Research" and "Claude Code," and leverage multi-LM architectures to overcome context limitations. Beneath their apparent diversity lies a recurring pattern: smaller "compressor" LMs (that can even run locally) distill raw context into compact text that is then consumed by larger "predictor" LMs. Despite their popularity, the design of *compressor-predictor* systems remains largely ad hoc, with little guidance on how compressor and predictor choices shape downstream performance. In practice, attributing gains to compression versus prediction requires costly, task-specific pairwise sweeps. We argue that these agentic system design questions are, at root, information-theoretic. Viewing the compressor LM as a *noisy channel*, we introduce a simple estimator of mutual information between the context and its compression to quantify compression quality in a task-independent way. We show that mutual information strongly predicts downstream performance, independent of any specific task. Through an information-theoretic framework, we perform a comprehensive empirical analysis across five datasets and three model families. Results reveal that larger compressors not only are more accurate, but also more token-efficient, conveying more bits of information per token. A 7B QWEN-2.5 compressor, for instance, is $1.6\times$ more accurate, $4.6\times$ more concise, and conveys $5.4\times$ more bits of mutual information per token than its 1.5B sibling. Across datasets, scaling compressors is substantially more effective than scaling predictors, enabling larger on-device compressors to pair with smaller cloud predictors. Applied to a Deep Research system, these principles enable local compressors as small as 3B parameters to recover 99% of frontier-LM accuracy at 26% of API costs.

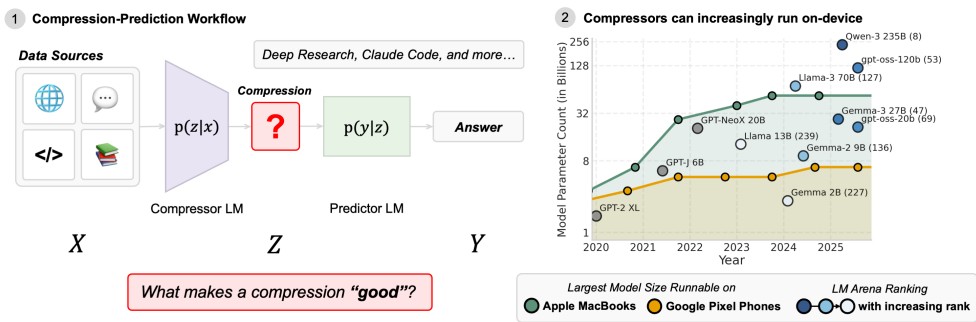

Figure 1: **Why compressors matter.** Many agentic LM systems rely on compressors, and personal devices are growing powerful enough to host them on-device. **(Left)** A compressor condenses a long input $X$ into a shorter summary $Z$ that a predictor expands into the final answer $Y$. **(Right)** Consumer hardware can now run increasingly large open-weight LMs, shown for Google Pixel phones and Apple MacBook laptops under FP16 precision with memory estimates from Modal (Lu, 2024). LM-Arena ranks indicate relative performance.

# 1 INTRODUCTION

Agentic language model (LM) systems have quickly become the backbone of modern AI workflows. From "Deep Research" systems (Hadfield et al., 2025) to Claude Code (Anthropic, 2025), millions of users now interact with pipelines where one model processes information and another builds on its outputs. Modern workflows commonly involve analyzing and generating more tokens than even the largest frontier models can handle effectively, degrading model performance—a failure mode referred to as *context rot* (Hong et al., 2025). Multi-LM systems coordinate multiple models to manage reasoning and memory beyond a single model's context window. While these architectures vary widely, a recurring pattern emerges across domains: smaller *compressor* models distill raw contexts into compact texts, which are then consumed by larger *predictor* models that output an answer and interact with the user (Figure 1) (Hadfield et al., 2025; Shen et al., 2023).

At present, however, designing *compressor-predictor* agentic systems remains largely trial-and-error. We lack a basic understanding of how the choice of compressor and predictor affects downstream performance. Specifically, we cannot determine whether credit belongs to the compressor's distillation or the predictor's reasoning—we lack task-agnostic methods to evaluate the compressor's outputs independently from downstream performance. This is because we are unable to measure how much of the original context the compressor actually preserves, which in turn determines how effectively the predictor can reason. This attribution problem has immediate practical consequences: as new models are released and practitioners swap components, they have no principled way to identify which module to improve without sweeping across the compound system from scratch.

To address this gap, we take an information-theoretic perspective, viewing the compressor as a *noisy channel* between the raw data and the predictor model. This framing allows us to evaluate communication between the two models rather than treat it heuristically. We propose using *mutual information* (MI) between the raw context and its compression as a task-agnostic measure of compressor efficacy—analogous to how perplexity serves as a task-agnostic predictor of downstream performance (Hoffmann et al., 2022; Kaplan et al., 2020). We then conduct a *rate-distortion analysis* to measure how downstream task performance varies with the degree of compression. While it is intractable to calculate MI between two token sequences linked via a nonlinear model, we develop a simple, unbiased estimator that can be computed via modern inference servers without requiring full vocabulary log probabilities.

With this new information-theoretic lens, we perform extensive empirical studies on five datasets (LONGHEALTH (Adams et al., 2024), FINANCEBENCH (Islam et al., 2023), QASPER (Dasigi et al., 2021), WILDCHAT (Zhao et al., 2024), and FINEWEB (Penedo et al., 2024)) to answer the following questions:

1. *Should you spend compute on the compressor or predictor?* We find that compressor quality overwhelmingly governs performance: scaling a QWEN-2.5 compressor from 1.5B to 7B improves accuracy by $60\%$ whereas scaling the predictor from 70B to 405B yields only a $12\%$ improvement on LONGHEALTH. This establishes a simple design principle: "front-load" compute into compressors, perhaps running on-device, to reduce dependence on massive cloud-hosted predictors. (Section 3.1)

2. *Which compressors are more token-efficient?* We find that larger compressors emit fewer output tokens while maintaining quality: in many model families, scaling compressor size not only improves accuracy but also produces compressions that are up to $4.6\times$ more concise. This token-efficiency yields sublinear scaling of FLOPs-per-generation as a function of model size. Strikingly, increasing QWEN-2.5 compressor from 1.5B to 7B, only adds $1.3\%$ more FLOPs-per-generation. (Section 3.1)

3. *Which factors determine compression quality and how do they relate to downstream performance?* We find that compressors' outputs carry up to $5.4\times$ more MI about the context (Section 3.2). Rate-distortion analysis reveals that information rate (MI per token) correlates strongly with downstream performance and perplexity ($r = -0.84$, $R^2 = 0.71$), providing a practical proxy for predicting system performance without full end-to-end evaluation (Section 3.3).

4. *With so many knobs to turn, which factors should you focus on for agentic system design?* We perform a meta-analysis across model families, sizes, and datasets, exposing a clear hierarchy of importance: compressor model family > compressor size > predictor size. (Section 3.4)

As a practical demonstration, we apply our findings to a simplified Deep Research pipeline, where a single predictor aggregates outputs from multiple compressors. This system achieves 99% of frontier-LM accuracy on the DEEPRESEARCH BENCH benchmark (Du et al., 2025) using local compressor models as small as 3B, reducing API costs by 74% (Section 3.5).

## 2 PRELIMINARIES

### 2.1 RELATED WORK

**Agents and Multi-Agent Systems**    We define AI agents as LMs that are embedded in a context and are able to reason, plan, and act through tool use (Belcak et al., 2025; Su et al., 2025; Wang et al., 2025; Yao et al., 2022). The agents are able to interact with its environment autonomously, enabling them to help us automate workflows and schedule our days (Pan et al., 2025). Modern workflows often require ingesting large corpora (enterprise documents, web search results). Relying on a single agent is both expensive in token cost and unreliable as context degrades ("context rot") (Hong et al., 2025; Zhang et al., 2024). It has become increasingly common to divide a complex task and context among multiple agents that take on different roles (Dang et al., 2025; Hadfield et al., 2025; Shen et al., 2023). Recent work has shown to improve cost efficiency by assigning smaller models to ingest the raw context while larger frontier models coordinate and orchestrate, effectively splitting the control plane and the data plane (Narayan et al., 2025).

Most prior work reports end-to-end utility such as accuracy, latency, and cost, while overlooking the communication channel itself. However, the use of information theory to explain and motivate the design of multi-agent systems has been limited, while there exists a large corpus of work on information theory in deep learning more generally. We analyze the intermediate communication, focusing on asymmetric compressor-predictor setups and their scaling.

**Information Theory in Deep Learning**    Information theory has been used to formally reason about intermediate representations in neural networks (Farquhar et al., 2024; Gardner, 1988; Kawaguchi et al., 2023; Morris et al., 2025). It has been used to provide notions of compression, information flow, hallucination, and model capacity. According to the information bottleneck principle, latent representations in graphical models trade-off input compression and downstream prediction ("compress as much as possible while being able to do the task") (Tishby et al., 2000). Prior work has shown that neural networks representations exhibit this trade-off, with trained layers naturally converging towards the information bottleneck minimum (Shwartz-Ziv & Tishby, 2017; Tishby & Zaslavsky, 2015). Other work attempts to understand the information dynamics of model learning and generalization (Goldfeld & Polyanskiy, 2020; Saxe et al., 2019; Westphal et al., 2025) by evaluating the information flow throughout training. Recent work leverages information theoretic quantities to both evaluate natural language generations and token sequences (Arda & Yener, 2025; Darrin et al., 2024; Fränken et al., 2024; Shani et al., 2025), and define training objectives and regularization terms (Chen et al., 2016; Hjelm et al., 2018; Kirsch et al., 2020; Oord et al., 2018; Wang et al., 2020). However, estimating information theoretic quantities such as MI in high dimensions is notoriously hard. Several works have shown that MI estimation can be highly sensitive to noise and sampling (Belghazi et al., 2018; McAllester & Stratos, 2020; Paninski, 2003).

### 2.2 INFORMATION THEORETIC PROBLEM SETUP

We extend the information theoretic framework beyond single-model analysis to study communication between two LMs.

We start with a simple compressor-predictor system, including one compressor LM, and one predictor LM. Let $X$ be the input context and $Y$ the target output. We consider a two-stage process

$$X \xrightarrow{p(z|x)} Z \xrightarrow{p(y|z)} Y.$$

The compressor $p(z \mid x)$ is modeled as a noisy channel, which compresses the context into a summary $Z$. The predictor $p(y \mid z)$ then uses this summary to generate the target output $Y$.

We proceed to define our MI estimator.

**Estimating mutual information** We want to measure the amount of information $Z$ contains about $X$, denoted as $I(X; Z)$. Larger MI values $I(X; Z)$ indicate that the compression retains more information about the original context. Mutual information estimation is a well-studied problem in statistics and machine learning (Murphy, 2023; Poole et al., 2019). However, many existing estimators are not practical in our setting. While many variational bounds require access to the underlying distribution or the training of an auxiliary model, we want to directly use the log probabilities exposed by LM inference engines.

To estimate MI, we start with the KL divergence (Kullback & Leibler, 1951) representation:

$$I(X; Z) = D_{\mathrm{KL}}\big(p(x, z) \,\|\, p(x)\, p(z)\big) = \mathbb{E}_{x, z \sim p(x, z)}\left[\log \frac{p(z|x)}{p(z)}\right].$$

While computing $p(z)$ is intractable, we can sample from and evaluate our encoder $p(z|x)$, and take samples from the data distribution $p(x)$,

$$I(X; Z) = \mathbb{E}_{x, z \sim p(x, z)}\left[\log \frac{p(z|x)}{\mathbb{E}_{x'}[p(z|x')]}\right],$$

$$\approx \frac{1}{NM} \sum_{i=1}^{N} \sum_{j=1}^{M}\left[\log p(z_{ij}|x_i) - \log\left(\frac{1}{N}\sum_{l=1}^{N} p(z_{ij}|x_l)\right)\right] \equiv \hat{I}(X; Z),$$

where $z_{ij} \sim p(z|x_i)$, $i = 1, \ldots, N$, $j = 1, \ldots, M$ and $x_l \sim p(x)$, $l = 1, \ldots, N$.

Note that $\hat{I}(X; Z) \leq \log(N)$, where the maximum is achieved when $p(z_{ij}|x_i) \gg p(z_{ij}|x_l)$ or $p(z_{ij}|x_i) \ll p(z_{ij}|x_l)$ for all $i, j$ and all $l \neq i$ (Appendix B.2). We do not claim theoretical optimality with this estimator. Instead, our aim is a practical estimator that can be computed directly using modern inference engines. With this estimator, we do not need access to the full probability distribution over the vocabulary, which allows us to use accelerated inference engines such as SGLang (Zheng et al., 2024).

In our experiments, we treat the raw context $X$ separately from a query $Q$ on that context. Thus, each compression $Z$ is generated conditioned on the query $Q$, so we estimate $I(X; Z \mid Q)$, which simply requires conditioning all terms on $Q$. While $I(X; Z \mid Q) \geq 0$, in practice, our Monte Carlo estimate can produce small negative values due to finite-sample variance. We correct these artifacts by clipping MI to zero. Furthermore, we find that LMs at 1–3B could assign high likelihoods to nonsensical token sequences, indicating miscalibration. Thus, we evaluated the log probabilities using a proxy model at the 7–8B scale. To mitigate biases, the proxy model was selected from a different family. We acknowledge that this could introduce variance and biases, and investigate their effects through ablation studies in Appendix E.1.4.

*Rate-distortion* theory quantifies the trade-off between *rate*—i.e., the amount of information the compression carries about the input—and *distortion*—the error in the prediction. We define *rate* (or *bit-efficiency*) as $R = \frac{I(X; Z|Q)}{L}$, for $L$ output tokens (measured in bits of mutual information per token). For simplicity, we define distortion as $D = 1 - \mathrm{ACC}(Z)$, by using the accuracy $0 \leq ACC(Z) \leq 1$. See the $R\text{-}D$ curve in Figure 19 (left). As rate increases, we expect distortion to converge towards a lower bound (irreducible error). See Appendix B.3 for further details.

## 3 RESULTS

We evaluate the compressor-predictor system as an information bottleneck across different tasks and domains. Beginning with a comprehensive scaling analysis of compressor and predictor model family and sizes, we find that larger compressors are not only more accurate but also more concise, leading to sublinear FLOPs-per-generation growth relative to model size. We conclude that scaling compressors is more effective than scaling predictors. Building on this, we show that mutual information rate closely tracks downstream accuracy and perplexity, providing a task-agnostic signal of compression quality. A meta-analysis highlights compressor model family and compressor size outweighing predictor size as the most important factors. Finally, we validate these principles in a Deep Research pipeline, where local compressors deliver frontier-level accuracy at a fraction of the cost.

**Datasets** We study our setup on five datasets, two synthetic, three non-synthetic: (a) LONGHEALTH, a set of synthetic clinical reports and 20 patient histories (Adams et al., 2024). We hide the multiple-choice options from the LMs to treat it as a question-answering (QA) dataset. (b) FINANCEBENCH, a collection of 150 10-K filings and reports paired with QA tasks (Islam et al., 2023). (c) QASPER, a QA dataset of 5,049 scientific research papers (Dasigi et al., 2021). (d) WILDCHAT, a large-scale LM conversation dataset (Zhao et al., 2024). (e) FINEWEB, a dataset of processed web pages from CommonCrawl (Penedo et al., 2024). See Appendix D.1 for more details on datasets.

**Evaluation procedure** We run each experiment with $S = 5$ random seeds. We evaluate prediction quality using accuracy for LONGHEALTH, FINANCEBENCH, and QASPER, assessing correctness of the predictions against the ground-truth using a GPT-4O-MINI judge. We use perplexity for WILDCHAT and FINEWEB, evaluating the log probabilities of a LLAMA-3.1-8B model.

**Compressor model** As compressors, we use smaller open-source LMs of the model families LLAMA-3 (Grattafiori et al., 2024), QWEN-2.5 (Qwen et al., 2025), and GEMMA-3 (Team et al., 2025). In our analysis, we choose to focus on GPT-style non-reasoning architectures. We additionally conduct preliminary experiments on reasoning and mixture-of-experts (MoE) LMs of the QWEN-3 family (Yang et al., 2025). See Appendix D.2 for further details on the compressor setup and Appendix C.1 for compressor prompts.

**Predictor model** We evaluate larger frontier models GPT-4O (OpenAI et al., 2024) as well as LLAMA-3 (1B, 8B, 70B, 405B) and QWEN-2.5 (72B) models as predictors. See Appendix C.2 for predictor prompts.

### 3.1 WHERE DO YOU NEED THE FLOPs: IN COMPRESSORS OR PREDICTORS?

We ask: should we scale compressors, which distill large amount of information into concise summaries, or predictors, which reason over the provided summaries to solve complex tasks?

In the following section, we show that scaling the compressor LM yields more significant gains. We vary the compressor model size and study its effects on downstream accuracy, the length of the compressed summaries, and the number of overall FLOPs-per-generation. We also establish scaling laws linking compressor model size to downstream performance.

First, we examine question-answering (QA) accuracy when scaling both compressor and predictor model size.

**Downstream performance is a function of compressor size.** We illustrate how downstream accuracy increases as model size increases (Figure 2, left). On LONGHEALTH, 7–8B models are up to $3.1\times$ more accurate than 1–1.5B models and surpass the GPT-4O-only baseline by 4pp. On FINANCEBENCH, 7–8B models are up to $2.6\times$ more accurate than 1–1.5B models and are able to recover 97% of the GPT-4O-only baseline. The same scaling behavior holds for GEMMA-3 models.

**Larger compressors are more concise.** In this section, we study the number of compression output tokens as a function of compressor size. We find that larger compressors are more concise (Figure 2, middle) without sacrificing accuracy. Specifically, 7–12B compressors are up to $4.6\times$ more token-efficient than their 1–1.5B counterparts within the same model family. QWEN-2.5 models tend to be more concise than LLAMA and GEMMA-3, suggesting that models can significantly vary in their communication profiles.

**Compression compute cost scales sublinearly with compressor size.** We combine the number of parameters with output token counts to estimate FLOPs-per-generation for each model family and size, i.e., the *actual* compute cost (Appendix B.1). Because larger compressors generate fewer tokens while maintaining accuracy, FLOPs-per-generation scale sublinearly with model size. Different model families exhibit distinct scaling behaviors (Figure 2, right): QWEN-2.5 compressors can scale from 1.5B to 7B parameters with only a 1.3% increase in FLOPs-per-generation on LONGHEALTH.

**Scaling compressors is more effective than scaling predictors.** Figure 3 shows that scaling the predictor LM provides only marginal improvements in accuracy once a baseline predictor capacity ($\approx$

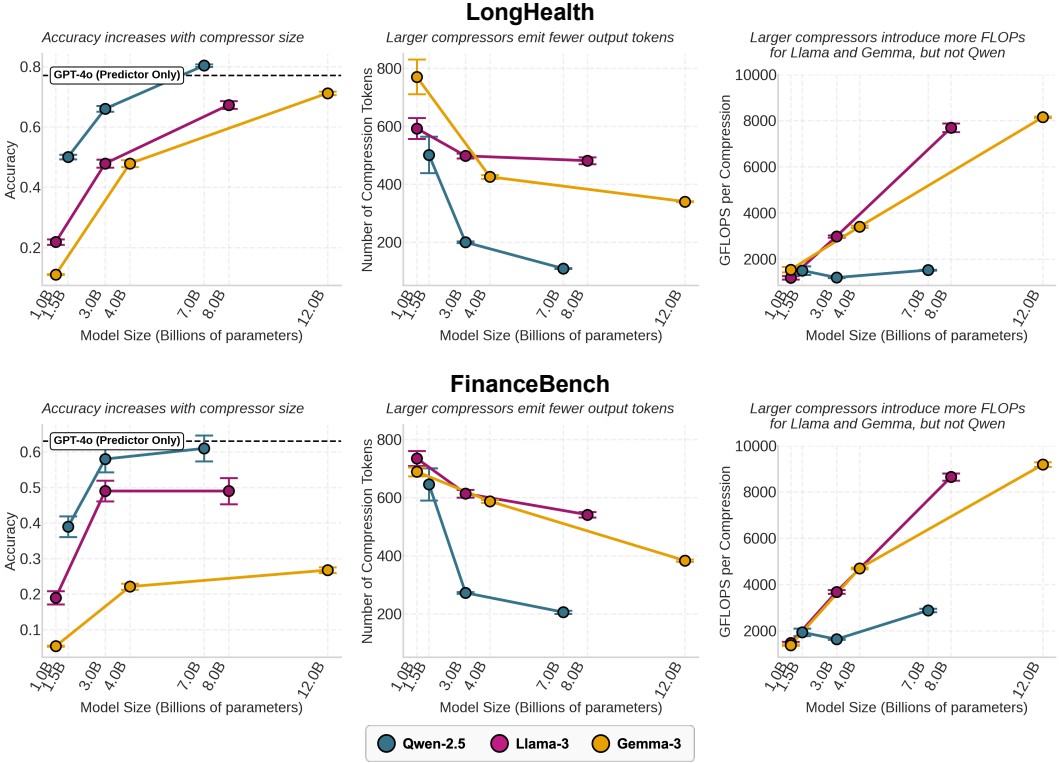

Figure 2: **Downstream accuracy, compression length, and compute cost scale with compressor size (Top: LONGHEALTH; Bottom: FINANCEBENCH).** We scale compressor model size and report a different metric on the $y$-axis of each column: **(Left)** accuracy, the dotted line being the GPT-4O baseline. **(Middle)** compression length, **(Right)** GFLOPs-per-compression. Vertical bars denote standard errors. Larger compressors produce shorter outputs with higher downstream accuracy. Similar trends hold on QASPER, WILDCHAT, and FINEWEB (Appendix E.1.1, E.1.2, E.1.3).

8B–70B) is reached. The gains in accuracy by increasing predictor size from 70B to 405B are within $12\%$ (LONGHEALTH) and $1\%$ (FINANCEBENCH). In contrast, scaling the compressor LM for both families leads to steeper increases in performance for fewer FLOPs-per-generation spent. For the QWEN-2.5 compressor family, FLOPs-per-generation meaningfully increase only when transitioning from 7B to 14B (models up to 7B all have roughly constant FLOPs-per-generation).

**You can trade local for remote compute.** As shown in Figure 1 (right), powerful models up to 27B can run without aggressive quantization on current-generation laptops. We anticipate the trends to continue and that even bigger models could run locally, and for free. Our results motivate "front-loading" FLOPs into local compressors to reduce cloud costs for serving the predictor (Figure 3).

**Analysis of compressor errors.** Errors in the compression step can be characterized into one of three categories: (a) the compression contains an incorrect answer ($36.3\%$ of compressor errors); (b) the compression contains no answer ($33.3\%$ of compressor errors); and (c) the compression omits details or parts of the information necessary for the answer ($30.4\%$ of compressor errors). For more details on compressor errors, refer to Appendix D.7.

## 3.2 WHICH COMPRESSORS MAXIMIZE COMMUNICATION EFFICIENCY?

We want to select compressors that provide maximal task-relevant information, ideally communicated in as few tokens as possible. Downstream QA accuracy and compression length do not fully capture compression quality. Instead, we turn to a information-theoretic framing: we estimate the mutual

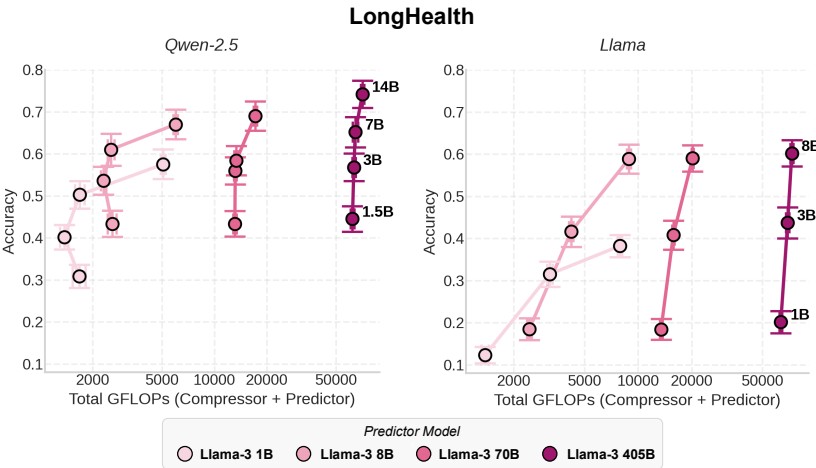

Figure 3: **Scaling compressors is more effective than scaling predictors on LONGHEALTH.** The $y$-axis reports accuracy and $x$-axis shows total compute cost in FLOPs-per-generation (log-scale). We compare compressor LMs from two families: **(Left)** QWEN-2.5, **(Right)** LLAMA-3. We scale predictor (color) and compressor (label) sizes and measure the total FLOPs-per-generation and downstream accuracy on QA tasks. Appendix E.1.6 shows consistent trends on FINANCEBENCH.

information $I(X; Z \mid Q)$ between context $X$ and generated compression $Z$ conditioned on query $Q$ for each compressor model in our scaling analysis, using the Monte Carlo estimator (Section 2.2).

**Larger compressors retain more mutual information and are more bit efficient.** We observe that $I(X; Z \mid Q)$ increases as compressor size increases (Figure 4). Larger, more expressive compressor models carry more mutual information between the original document and the compressed summary.

On LONGHEALTH, while LLAMA compressors are far from the theoretical maximum, we find that QWEN-2.5 and GEMMA-3 models produce compressions that saturate in mutual information at the largest model sizes. By contrast, on FINANCEBENCH, mutual information saturates already at the 3B scale. We observe this saturation behavior primarily on datasets with a highly heterogeneous corpus of context documents.

Combining the scaling effects of mutual information with the observation that larger compressors omit fewer tokens, we find that larger compressors are more bit efficient (Figure 4).

We ablate across multiple proxy model choices and find that the mutual information scaling trends remain consistent across different proxy choices (Appendix B.3). Furthermore, we estimate mutual information without a proxy model for compressors of the QWEN-3 family and observe the same scaling behavior (Appendix E.1.5).

**Compressor scaling effects are consistent across prompt conditions.** A natural concern is whether our scaling results depend on specific prompt formatting. To test robustness, we instructed compressor models to output 3, 6, or 9 sentences, varying conciseness levels. Scaling behavior in accuracy, compression output size, FLOPs-per-generation, MI, and bit efficiency remained consistent across all conciseness instructions on both LONGHEALTH and FINANCEBENCH (Figures 5, 16). The relative improvements from larger compressors persist regardless of prompted compression output length, confirming that model capacity drives the observed efficiency gains.

### 3.3 INFORMATION RATE CORRELATES STRONGLY WITH DOWNSTREAM PERFORMANCE

**Mutual information and bit-efficiency are proxies for system performance.** Information rate (bit-efficiency) is closely related to distortion $(1 - \text{accuracy})$. Motivated by the classical form of the rate-distortion function for a independent Gaussian source $X$, we fit decaying exponential functions to the rate-distortion data (Appendix B.3). This fit characterizes the correlation between information

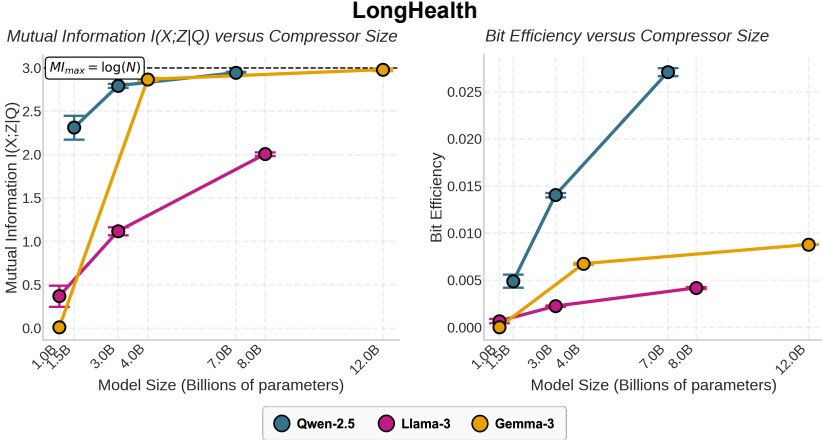

Figure 4: **Larger compressors generate outputs that carry more information about their inputs (conditioned on the query) on LONGHEALTH.** We scale compressor model size and estimate the **(Left)** mutual information, and **(Right)** bit efficiency (bits of mutual information per token; higher is better) carried by their outputs. Larger compressor model sizes compress documents with higher mutual information and bit efficiency. The dotted line represents the theoretical maximum of the mutual information estimator at the natural logarithm $\log(N)$, where $N$ is the number of documents mutual information is computed across. We find consistent trends on FINANCEBENCH and QASPER (Appendix E.1.6, E.1.1).

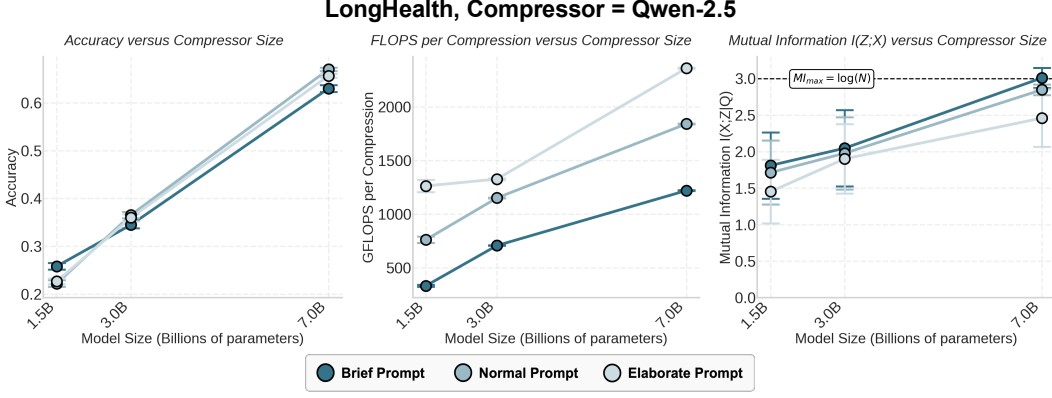

Figure 5: **Scaling behavior of compressor model size hold across instructed conciseness (COMPRESSOR = QWEN-2.5).** We ablate over compression conciseness levels by varying the compression prompt instructions. We measure **(Left)** accuracy, **(Middle)** GFLOPs-per-generation, and estimate **(Right)** mutual information. We find that accuracy and mutual information are largely unaffected by conciseness instructions. Compressors instructed to be more concise are more token-efficient, and thus compute-efficient. Scaling trends in accuracy, compute cost, and mutual information hold across conciseness constraints. Appendix E.1.7 shows analogous results on FINANCEBENCH.

rate and distortion and corroborates our previous finding that scaling predictors beyond 70B yields only marginal improvements in distortion (Figure 6, left).

Furthermore, Figure 6 (right) reveals that mutual information is also strongly correlated with perplexity ($r = -0.84$, $R^2 = 0.71$) for extractive tasks on FINEWEB (detailed in Appendix D.1.5).

**Predictors do not prefer compressors of the same family.** Further rate-distortion analysis across QWEN-2.5 and LLAMA reveals that distortion is primarily dependent on model family and size. Crucially, predictors do not perform better when paired with compressors of the same family (Figure 19).

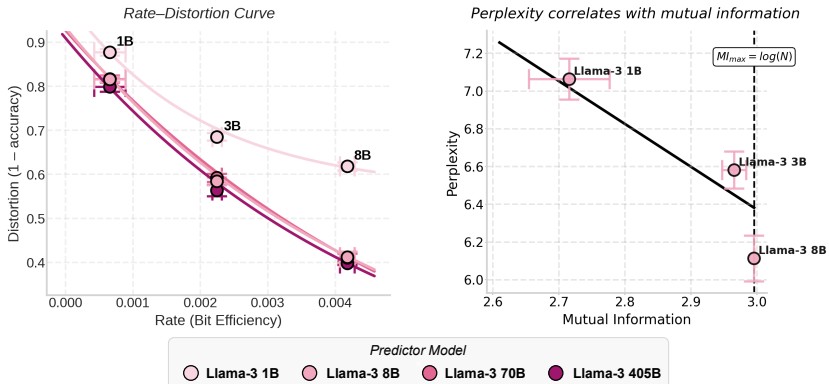

Figure 6: **Mutual information and bit efficiency correlate strongly with downstream performance. (Left)** We vary both predictor and compressor model in the compression-prediction workflow and measure the distortion on the $y$-axis and estimate the rate on the $x$-axis. We plot the resulting rate-distortion curves across predictor sizes 1B, 8B, 70B, and 405B for LLAMA compressors on LONGHEALTH. The lines show fitted exponential-decay functions. **(Right)** We measure perplexity and mutual information on compressions generated by LLAMA compressors on FINEWEB. The line shows a fitted linear function ($r = -0.84$, $R^2 = 0.71$). Appendix E.2 provides further analyses.

### 3.4 WHICH KNOBS TO TURN?

To guide practical system design, we analyze which components of the compression-prediction pipeline most strongly drive downstream QA accuracy. We fit a logistic regression predicting binary correctness on LONGHEALTH and FINANCEBENCH using the features specified in Appendix D.4. The compressors we consider are QWEN-2.5 and LLAMA models and predictors are LLAMA models of sizes 1B, 8B, 70B, and 405B.

Our analysis reveals that compressor model family is the most important factor (Figure 17) with QWEN-2.5 compressors outperforming LLAMA. Additionally, scaling the compressor LM matters substantially more than scaling the predictor LM, confirming previous findings in Section 3.1.

### 3.5 SCALING DEEP RESEARCH

We evaluate our compression-prediction framework on open-domain "Deep Research" workflows, where a predictor LM decomposes research tasks into subtasks and aggregates compressor outputs into final reports. We use DEEPRESEARCH BENCH (Du et al., 2025), which assesses system performance across four dimensions: Comprehensiveness, Depth, Instruction-following, and Readability. These four dimensions form a quantitative *RACE* (Reference-based Adaptive Criteria Evaluation) score. We measure cost per task based on current API prices (as of August 2025), which are time-sensitive, but serve as good approximations for relative cost differences between models. We vary predictor sizes across the LLAMA family and compressor sizes across the QWEN-2.5 family. LM Prompts are detailed in Appendix C.3 and full experimental details are in Appendix D.6.

Larger predictor models consistently improve *RACE* scores, while larger compressors provide substantial performance gains at minimal additional API costs (Figure 7).

As a baseline, we evaluate the results of providing uncompressed web search data to a GPT-4O predictor. A QWEN-2.5-14B compressor paired with a GPT-4O predictor achieves 2.3% higher RACE scores at only 28.1% of the API cost compared to the uncompressed baseline. We detail further findings in our scaling experiments in Appendix E.3.

## 4 DISCUSSION

We establish an information-theoretic framework for compressor-predictor systems to determine how model selection and scaling affect compute efficiency. Our findings come with important

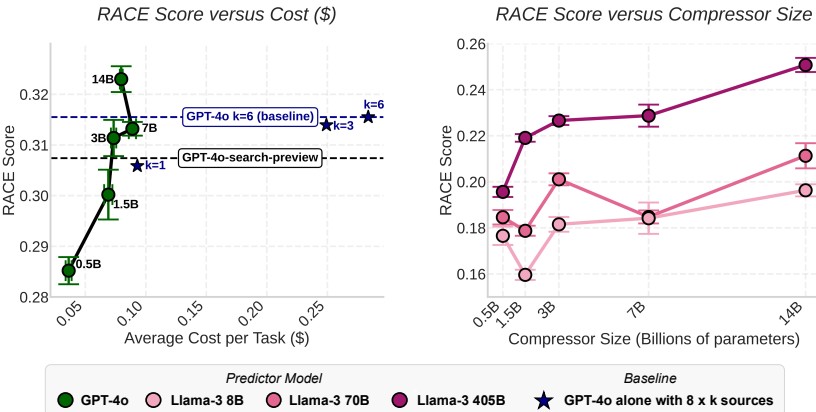

Figure 7: **Deep Research Scaling Results. (Left)** RACE score versus average task cost when using GPT-4O as a predictor with QWEN-2.5 compressors of varying sizes. Costs are based on GPT-4O API rates (August 2025: \$2.50/1M input tokens, \$10.00/1M output tokens). Larger compressors improve performance with minimal cost increases. For reference, we include GPT-4O results without compression and also for the GPT-4O-SEARCH-PREVIEW model. **(Right)** RACE scores for different QWEN-2.5 compressor sizes (0.5B–14B) under three LLAMA predictors (8B, 70B, 405B).

limitations: at the 1–3B model scale, our MI estimator relies on proxy models and log probabilities, introducing potential variance and biases. Furthermore, we primarily focus on GPT-style non-reasoning models with single-round communication, limiting generalizability to reasoning-augmented models or iterative multi-agent workflows. While we provide initial results for reasoning and mixture-of-experts models, future work should extend the information-theoretic analysis to a wider range of model families and evaluate reasoning traces in a more principled manner.

Several research directions warrant investigation. Mutual information estimation for LM outputs remains challenging, though alternative estimators like INFONCE (Aitchison & Ganev, 2021) offer promising solutions. Information-theoretic principles could guide compressor routing strategies and fallback decisions for remote full-context processing. Training objectives based on rate-distortion analysis represent another avenue to optimize compressor-predictor communication. While we define compression as summarization in this work, compression can also be found in other agentic-system workflows, such as structured extraction and function-call generation. FLOPs-per-generation is an intuitive measurement of compute cost. However, device-specific efficiency optimizations are crucial in real-world deployment settings and warrant further analysis. Finally, mixture-of-experts (MoE) models (Fedus et al., 2022) may exhibit different scaling behaviors since their compute cost depends on activated experts rather than total parameter count.

Overall, we distill our findings into four principles for agentic system design:

> **Principles for Agentic System Design**
>
> - **Compressors can be scaled at a sublinear computational cost.** Since larger models are more information-efficient (emit fewer tokens with higher information-density), FLOPs-per-generation scale sublinearly as a function of model size.
> - **"Front-load" compute into local compressors to reduce remote costs.** Scaling compressors is more effective than scaling predictors. By running larger compressors on-device, we can reduce predictor serving costs in the cloud.
> - **Optimize for information density.** The mutual information between an input context and an agent output is a task-agnostic indicator of compression quality and is tightly linked to downstream performance and perplexity.
> - **Expect model family to differ in scaling trends.** Choice of compressor and predictor model family yields offsets in rate-distortion curves and scaling effects. QWEN-2.5 compressors scale more compute-efficiently than LLAMA and GEMMA-3. QWEN-2.5 predictors yield higher accuracies than LLAMA.

ACKNOWLEDGMENTS

We thank Simran Arora, Yasa Baig, Kelly Buchanan, Sabri Eyuboglu, Neel Guha, Simon Guo, Jerry Liu, Jack Morris, Simon Pritchard, Jon Saad-Falcon, Frederic Sala, Ravid Shwartz-Ziv, Seonghyun Yoon, and the entire Linderman and Hazy research labs for helpful discussions and feedback. We are incredibly grateful to Modal and Together AI for providing the GPUs to support this work.

We gratefully acknowledge the support of NIH under No. U54EB020405 (Mobilize), NSF under Nos. CCF2247015 (Hardware-Aware), CCF1763315 (Beyond Sparsity), CCF1563078 (Volume to Velocity), and 1937301 (RTML); US DEVCOM ARL under Nos. W911NF-23-2-0184 (Long-context) and W911NF-21-2-0251 (Interactive Human-AI Teaming); ONR under Nos. N000142312633 (Deep Signal Processing); Stanford HAI under No. 247183; Google DeepMind; Google Research; Google Cloud; NXP, Xilinx, LETI-CEA, Intel, IBM, Microsoft, NEC, Toshiba, TSMC, ARM, Hitachi, BASF, Accenture, Ericsson, Qualcomm, Analog Devices, Salesforce, Total, the Laude Institute, Prime Intellect, Anthropic, the HAI-GCP Cloud Credits for Research program, the Stanford Data Science Initiative (SDSI), and members of the Stanford DAWN project: Meta, Google, and VMWare; and members of the Stanford SEAMS project: IBM and Felicis. The U.S. Government is authorized to reproduce and distribute reprints for Governmental purposes notwithstanding any copyright notation thereon. Any opinions, findings, and conclusions or recommendations expressed in this material are those of the authors and do not necessarily reflect the views, policies, or endorsements, either expressed or implied, of NIH, ONR, or the U.S. Government.

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

We utilized AI tools to assist with code implementation and manuscript proofreading.

ETHICS STATEMENT

Regarding fairness and accessibility, our recommendation to "front-load" computation into local compressors may create barriers for researchers with limited hardware resources, potentially exacerbating inequalities in AI access despite reducing cloud API costs by 74%. The compression techniques process documents through multiple model stages, raising privacy concerns about information retention in compressed representations, especially when handling sensitive data. Our efficiency improvements could accelerate broader deployment of agentic systems with both beneficial and harmful applications, while the environmental impact of encouraging larger local model deployment (up to 27B parameters) requires careful consideration against potential increases in aggregate energy consumption. We encourage practitioners to implement appropriate privacy safeguards and consider the dual-use implications of these compression-prediction architectures as they become more prevalent.

REPRODUCIBILITY STATEMENT

We provide comprehensive implementation details and experimental specifications throughout the paper and appendices. Section 2.2 contains the complete derivation and implementation of our mutual information estimator, while Appendix B.1 details the FLOPs computation methodology for dense transformer models. All experimental configurations, including model selections, hyperparameters, and prompt templates, are specified in Appendices B and D. Dataset construction procedures are documented in Appendix D.1, with specific sampling criteria for each of the four datasets, as well as prompt templates to construct synthetic QA and generation tasks. All models used to generate synthetic tasks and answers are detailed in Appendix D.1. The Deep Research experimental setup is fully described in Appendix D.6, including the complete workflow implementation and evaluation framework, as well as prompt templates for compressor and predictor LMs. All experiments run on $S = 5$ random seeds with reported standard errors, and we specify the exact model versions, inference parameters, and evaluation protocols used across all experiments. Rate-distortion analysis parameters and fitting procedures are detailed in Appendix B.3, while the generalized linear model specifications for meta-analysis are provided in Appendix D.2.

APPENDIX

## A  EXTENDED RELATED WORK

**Agents and Multi-Agent Systems**  Multi-Agent system design choices studied in the literature include model size (Wang et al., 2024), number of agents and communication rounds (Chen et al., 2024; Kim et al., 2025; Schluntz & Zhang, 2025), research depth (Zhang et al., 2025b), agentic state and communication (Li et al., 2025; Zhang et al., 2025a; Zou et al., 2025), and planning or decomposition strategies (Chen et al., 2023; Erdogan et al., 2025; Yao et al., 2023). Overall, they find that performance improves with larger models, more agents, additional rounds, and increased task-relevant context. Other research focuses on learned or automated optimization of agentic architectures (Hu et al., 2024; Jiang et al., 2025; Saad-Falcon et al., 2024) and agentic memory systems (Han et al., 2025; Zhong et al., 2024; Zhou et al., 2025).

**Deep Research**  Over the past year, large-scale Deep Research systems have been popularized and adopted by frontier labs in industry such as OpenAI, Anthropic, and xAI. These systems commonly have an asymmetric setup, where a high-capacity *predictor* LM decomposes the user query into subtasks that are executed by *compressor* models in parallel (Schluntz & Zhang, 2025). The results of these subtasks are synthesized into one answer that is commonly presented to the user as a comprehensive *research report*. In practice, compressor models can range from frontier models (Hadfield et al., 2025; Schluntz & Zhang, 2025) to local small LMs (Narayan et al., 2025). Recent works have been centered around establishing evaluation benchmarks to compare different agentic *Deep Research* systems (Du et al., 2025; Guha et al., 2023; Nakano et al., 2021). These benchmarks focus on measuring the output quality and downstream utility of generated research reports.

## B  EXTENDED DESCRIPTION OF METHODS

In this section, we provide a more in-depth explanation and derivation of our information-theoretic approach and problem setup.

### B.1  COMPUTE COST OF DENSE LMS

We measure compute cost of each compressor/predictor LM call by the number of FLOPs per token in each forward pass through our dense transformer-based LMs as

$$C_{\text{dense}} \approx 2N_{\text{params}} + 2n_{\text{layer}}n_{\text{ctx}}d_{\text{attn}},$$

with model size $N_{\text{params}}$, number of input context tokens $n_{\text{ctx}}$, number of layers $n_{\text{layer}}$, and number of attention heads per layer $d_{\text{attn}}$ (Kaplan et al., 2020). We observe that FLOPs-per-token-generated for dense models grows roughly linearly with model size.

### B.2  THEORETICAL ANALYSIS: BOUNDS OF MONTE CARLO ESTIMATOR

The Monte-Carlo estimator of mutual information between X and Z is upper bounded by

$$\hat{I}(X; Z) \leq \log N,$$

where $N$ is defined as the number of contexts sampled from X.

*Proof.* Consider the Monte-Carlo estimator of mutual information

$$\hat{I}(X; Z) = \frac{1}{NM} \sum_{i=1}^{N} \sum_{j=1}^{M} \left[ \log p(z_{ij}|x_i) - \log \left( \frac{1}{N} \sum_{l=1}^{N} p(z_{ij}|x_l) \right) \right]$$

$$= \frac{1}{NM} \sum_{i=1}^{N} \sum_{j=1}^{M} \left[ \log p(z_{ij}|x_i) + \log N - \log \left( \sum_{l=1}^{N} p(z_{ij}|x_l) \right) \right].$$

For any fixed summary $z_{ij}$ and context $x_l$, we have

$$\sum_{l=1}^{N} p(z_{ij}|x_l) \geq \max_l \ p(z_{ij}|x_l)$$

$$\geq p(z_{ij}|x_i).$$

Plugging this into the estimator bounds yields the upper bound

$$\hat{I}(X;Z) = \frac{1}{NM} \sum_{i=1}^{N} \sum_{j=1}^{M} \left[ \log p(z_{ij}|x_i) + \log N - \log \left( \sum_{l=1}^{N} p(z_{ij}|x_l) \right) \right]$$

$$\leq \log N.$$

To see when the bound is tight at $p(z_{ij}|x_i) \gg p(z_{ij}|x_l) \ \forall l \neq i$, we write the denominator using a constant $c_{ij}$

$$\sum_{l=1}^{N} p(z_{ij}|x_l) = e^{c_{ij}} \sum_{l=1}^{N} e^{\log p(z_{ij}|x_l) - c_{ij}}.$$

Choose $c_{ij} = \log p(z_{ij}|x_i)$, this gives us

$$\hat{I}(X;Z) = \frac{1}{NM} \sum_{i=1}^{N} \sum_{j=1}^{M} \left[ \log p(z_{ij}|x_i) + \log N - c_{ij} - \log \left( \sum_{l=1}^{N} e^{\log p(z_{ij}|x_l) - c_{ij}} \right) \right]$$

$$= \log N - \frac{1}{NM} \sum_{i=1}^{N} \sum_{j=1}^{M} \left[ \log \left( \sum_{l=1}^{N} e^{\log p(z_{ij}|x_l) - \log p(z_{ij}|x_i)} \right) \right],$$

Since $p(z_{ij}|x_i) \gg p(z_{ij}|x_l), \forall l \neq i$,

$$\hat{I}(X;Z) \approx \log N + \frac{1}{NM} \sum_{i=1}^{N} \sum_{j=1}^{M} \left[ -\log(1 + \varepsilon) \right] \qquad (\varepsilon \approx 0)$$

$$\approx \log N.$$

Thus $\log N$ is a tight upper bound. $\qquad\qquad\qquad\qquad\qquad\qquad\qquad\qquad\qquad\qquad\qquad$ $\square$

### B.3 RATE-DISTORTION-THEORY

Assume $X$ to be an independent Gaussian random variable with variance $\sigma^2(X)$, the canonical rate-distortion function is

$$R(D) = \begin{cases} \frac{1}{2} \log\left(\frac{\sigma^2(X)}{D}\right), & 0 \leq D \leq \sigma^2(X) \\ 0, & D > \sigma^2(X). \end{cases}$$

We illustrate rate-distortion curves as distortion $D$ (how much accuracy is lost in communication) versus rate $R$ (how many bits spent encoding data). Inverting the expression for the rate-distortion function gives (Cover & Thomas, 2005)

$$D_{Gaussian}(R) = \sigma^2 \, 2^{-2R}$$

$$= \sigma^2 \, e^{-2\ln 2 \, R}$$

$$= C \, e^{-bR}, \qquad \text{with} \quad C = \sigma^2, \ b = 2\ln(2).$$

The Gaussian rate-distortion function does not describe LM data distributions. However, it serves as a closed-form model that captures the qualitative shape of typical rate-distortion curves. We use the Gaussian rate-distortion function as a simplified model to qualitatively compare different predictors.

In practice, we treat $C$ and $b$ as function parameters to account for unknown variance and modeling noise. LM compression-prediction systems often exhibit a non-zero distortion floor (e.g. imperfect LM judge, label noise, predictor expressive power), which we account for through offset $D_0$. $D_0$ is a lower bound of the distortion in the system as rate (bit efficiency) increases,

$$D(R) = C \, e^{-bR} + D_0.$$

We fit exponential decay functions to the rate-distortion curves based on the least-squares estimates $(\hat{C}, \hat{b}, \hat{D}_0)$.

# C    PROMPTS

## C.1    COMPRESSOR MODEL PROMPTS

We use the following prompt templates to compress the raw context documents on LONGHEALTH, FINANCEBENCH, QASPER, FINEWEB, and each chat conversation on WILDCHAT:

---

**Query-Specific Base Compression Prompt Template**

```
Summarize the following text to include ONLY information needed to answer the question.
Extract the key points relevant to the question.
DO NOT ANSWER THE QUESTION DIRECTLY.

Question:
{query}

Text:
{text}

Your summary (make sure to include all important details / background information related to the
*question*.  **DO NOT ANSWER THE QUESTION**)
```

---

**Memory Construction/Compression Prompt Template (WildChat)**

```
You are a memory compression assistant, tasked with summarizing a chat conversation.
Produce a summary that preserves all details that could be useful as memory for a language model.  DO NOT
invent any information.

CHAT:
{conversation}

Your summary (Just plain text, no formatting.)
```

---

**Query-Agnostic Compression Prompt Template (FineWeb)**

```
Summarize the following text and produce a summary that preserves all details that could be needed to
answer likely questions about the text.  Do NOT invent facts.

Do NOT answer any question; just summarize potential answer-bearing info.

Text:
{text}

Your summary (make sure to include all important details / background information related.  Just plain
text, no formatting.)
```

---

## C.2    PREDICTOR MODEL PROMPTS

Given the compressor output, we answer extractive QA tasks on LONGHEALTH, FINANCEBENCH, QASPER, WILDCHAT, and FINEWEB, and creative tasks on FINEWEB using the following prompt templates:

---

**Base Prediction Prompt Template**

```
Please answer the following question based on the provided summary.
Question:
{query}

Summary:
{summary}

Please respond in the following JSON format:  <briefly think about the information you have and the
question you need to answer>

{{
    "explanation":  "<brief explanation of the answer.  explain how you arrived at the answer.  1-2
sentences>",
    "answer":  "<your final answer>"
}}

Your answer (YOU MUST ONLY RESPOND WITH THE JSON OBJECT):
```

---

**WildChat Prediction Prompt Template**

```
Please answer the following question based on the provided chat memory.

Question:
{query}
```

---

```
Memory:
{memory}

Please respond in the following JSON format:  <briefly think about the information you have and the
question you need to answer>

{{
    "answer":  "<your final answer>"
}}

Your answer (YOU MUST ONLY RESPOND WITH THE JSON OBJECT):
```

**FineWeb Prediction Prompt Template (Extractive)**

```
Please answer the following question based on the provided {context_type}.

Question:
{query}

{context_type}:
{summary}

Please respond in the following JSON format:
<briefly think about the information you have and the question you need to answer>

{{
    "answer":  "<your final answer>"
}}

Your answer (YOU MUST ONLY RESPOND WITH THE JSON OBJECT):
```

**FineWeb Prediction Prompt Template (Creative)**

```
Please do the following based on the provided {context_type}.

Task:  {query}

{context_type}:
{summary}

Please respond in the following JSON format:
<briefly think about the information you have and the question you need to answer>

{{
    "answer":  "<your final answer>"
}}

Your answer (YOU MUST ONLY RESPOND WITH THE JSON OBJECT):
```

## C.3   DEEPRESEARCH PROMPTS

The following prompt templates were used sequentially as the backbone for our compressor-predictor Deep Research workflow.

**DeepResearch Query Generation Prompt Template (Predictor)**

```
You are a research supervisor tasked with comprehensively exploring a research topic.  Use a strategic,
top-down approach to design your research.

Research Topic:  {query}

**PHASE 1:  RESEARCH PLANNING**
First, analyze this research topic and create a comprehensive research plan.  Consider:
- What are the key areas that must be investigated to fully understand this topic?
- What specific objectives will guide your research?
- How do different aspects of this topic relate to each other?
- What types of information will be most valuable for a complete analysis?
- What is the logical flow for presenting findings?

**PHASE 2:  STRATEGIC QUERY GENERATION**
Based on your research plan, generate EXACTLY 8 different search queries that together will provide
comprehensive coverage of this topic.  Each query should serve a specific strategic purpose in your
overall research architecture.

For each search query, provide a specific sub-task/question that explains how it serves your research
plan.

Return your response in this exact JSON format:
{{
    "research_plan":  "Your comprehensive research architecture and strategic objectives for investigating
this topic.  Explain the key areas to investigate, how they relate, and the logical structure for
analysis.",
    "queries":  [
        {{
```

```
            "search_query":  "specific search terms optimized for Google",
            "sub_task":  "What specific question does this query address and how does it serve the research
plan?"
      }},
      {{
            "search_query":  "second strategic search query",
            "sub_task":  "What does this query aim to discover and how does it fit the research
   architecture?"
      }},
      {{
            "search_query":  "third targeted search query",
            "sub_task":  "What aspect does this explore and why is it essential to the research plan?"
      }},
      {{
            "search_query":  "fourth strategic search query",
            "sub_task":  "What question does this answer and how does it complement other queries?"
      }},
      {{
            "search_query":  "fifth focused search query",
            "sub_task":  "What aspect does this cover and how does it build on previous queries?"
      }},
      {{
            "search_query":  "sixth comprehensive search query",
            "sub_task":  "What additional dimension does this explore and why is it crucial?"
      }},
      {{
            "search_query":  "seventh strategic search query",
            "sub_task":  "What specific gap does this fill in the research architecture?"
      }},
      {{
            "search_query":  "eighth concluding search query",
            "sub_task":  "What final aspect does this cover and how does it complete the comprehensive
research?"
      }}
   ],
   "synthesis_strategy":  "Detailed strategy for combining findings from all 8 queries based on your
research plan.  Explain how the information will be structured, what relationships will be highlighted,
and how the final analysis will be organized to maximize comprehensiveness and insight."
}}

**Strategic Guidelines:**
1.  Each search query should be 3-8 well-chosen keywords targeted for your specific research objectives
2.  Design queries to serve complementary roles in your research architecture (not just generic
dimensions)
3.  Ensure queries are strategically coordinated to provide comprehensive topic coverage
4.  Each sub-task should explain how the query serves your overall research plan
5.  Create a synthesis strategy that reflects your planned research structure

**Research Focus Areas to Consider:**
- Foundational understanding and current state
- Key challenges, problems, or limitations
- Solutions, methodologies, and best practices
- Evidence, data, and empirical findings
- Future trends, developments, and implications
- Multiple perspectives and stakeholder viewpoints

CRITICAL: You must return ONLY the JSON object.  Do NOT format it as a code block with ```json``` or
any other markdown formatting.  Return the raw JSON object directly.
```

## DeepResearch Synthesis Prompt Template (Predictor)

```
You are tasked with creating a comprehensive, high-quality research report for a DeepResearch task.  You
have extensive research findings below - use ALL of them to create a detailed, thorough analysis.

**Original Research Task:** {original_task}

**Research Plan:** {research_plan}

**Research Findings:**
{qa_pairs}

**Synthesis Strategy:** {synthesis_strategy}

**COMPREHENSIVE INFORMATION UTILIZATION - ALL SOURCES REQUIRED:**
You must systematically work through ALL the provided research findings above.  Do not selectively use
only some information - your report must demonstrate that you have reviewed and integrated ALL relevant
details, data points, examples, and perspectives from every query and source provided.

**REPORT STRUCTURE AND REQUIREMENTS:**
1.  **Detailed Background Context** - Provide extensive background and context
2.  **Comprehensive Analysis** - Multiple detailed sections covering all aspects
3.  **Extensive Evidence Integration** - Use specific examples, data, quotes from ALL sources
4.  **Thorough Implications Discussion** - Detailed analysis of implications and significance
5.  **Complete Conclusions** - Comprehensive conclusions and future research directions

**WRITING REQUIREMENTS FOR HIGH QUALITY:**
- Write detailed explanations, not brief summaries
- Include extensive examples and case studies from the research
- Provide comprehensive background and context for every major point
- Use all statistical data, quotes, and specific details from the research findings
```

```
- Elaborate on implications, significance, and broader connections
- Include detailed analysis of methodologies, approaches, and frameworks mentioned
- Discuss limitations, challenges, and areas for further research extensively

Create a thorough academic research report that:
- Uses extensive detail and comprehensive analysis throughout
- Integrates ALL findings with detailed explanations and context
- Provides comprehensive coverage with extensive supporting evidence
- Includes detailed discussion of all relevant aspects and implications
- Demonstrates mastery of the subject through thorough, detailed analysis

**FINAL REQUIREMENT:**
Your response must be substantial and comprehensive. Write extensively with exhaustive detail,
comprehensive analysis, and complete utilization of all research findings. Provide truly comprehensive
coverage of the topic that demonstrates thorough understanding and integration of all available research.
```

## DeepResearch Source Summarization Prompt Template (Compressor)

```
Your job is to extract detailed, specific information from the following content to support comprehensive
research analysis.

**Main Research Query:** {query}

**Specific Sub-task/Question:** {sub_task}

## Content
{content}

**EXTRACTION REQUIREMENTS: Provide a detailed and comprehensive extraction that captures:**

**Factual Information:**
- Specific numbers, statistics, percentages, and quantitative data
- Dates, timelines, and chronological information
- Names of people, organizations, companies, and institutions
- Geographic locations, regions, and jurisdictions
- Technical specifications, measurements, and benchmarks

**Detailed Examples and Evidence:**
- Concrete case studies and real-world examples
- Specific research findings and study results
- Direct quotes and expert opinions
- Policy details and regulatory information
- Implementation details and methodologies

**Comprehensive Coverage:**
- Key facts directly relevant to both the main query AND the specific sub-task
- Important concepts, definitions, and explanations
- Cause-and-effect relationships and underlying mechanisms
- Trends, patterns, and developments over time
- Challenges, limitations, and problem areas identified

**Analytical Insights:**
- Implications and significance of the information
- Relationships between different data points
- Comparative information and benchmarks
- Future projections and forecasted trends
- Expert assessments and professional evaluations

Focus on depth and specificity while maintaining clarity. Extract comprehensive, specific information
with extensive detail, numbers, examples, and evidence. Do not provide brief summaries - ensure your
extraction is thorough and substantial. Extract information that would be valuable for creating a
comprehensive research report. Pay special attention to information that directly addresses the sub-task
question.

Return your extraction in JSON format with these fields:
- "explanation": Your detailed extraction of specific information, facts, data, examples, and evidence
with extensive detail
- "answer": "relevant" if this content contains information relevant to the query and sub-task, "not
relevant" otherwise

CRITICAL JSON FORMATTING RULES:
- Replace all double quotes (") inside text with single quotes (')
- Replace all newlines with spaces
- Ensure the JSON is valid and parseable
- Do NOT use line breaks within the JSON fields

Example format:
{{"explanation": "Your detailed extraction with specific facts, numbers, examples, and evidence using
single quotes for any nested quotes", "answer": "relevant"}}

CRITICAL: You must return ONLY the JSON object. Do NOT format it as a code block with ```json``` or
any other markdown formatting. Return the raw JSON object directly.
```

## D    EXTENDED EXPERIMENTAL SETUP

Here, we further explain the construction of our datasets, choice of compressor and predictor models, and Deep Research experimental setup.

### D.1    DATASETS

#### D.1.1    LONGHEALTH

LONGHEALTH is a QA benchmark composed of 20 patient cases and clinical documents. Each of the 20 patients has a set of 20 multiple-choice questions about their personal records each ranging from 5,090 to 6,754 words (Adams et al., 2024). The original LONGHEALTH benchmark is a multiple-choice QA task. To more closely mirror our QA setups in the remaining three datasets, we remove the multiple-choice options in the prediction step. We subsample $N = 20$ documents and queries and generate $M = 20$ compressions for each of the problem contexts.

#### D.1.2    FINANCEBENCH

FINANCEBENCH is a long-context QA benchmark on 150 financial reports. Each financial report ranges from 1,923 to 517,224 tokens, with an average length of 119,968 tokens (Islam et al., 2023). We filter the original FINANCEBENCH dataset to only include samples with answer evidence at one location in the text. We slice a text segment of 21,500 tokens centered around the evidence as the raw document context. We subsample $N = 20$ problems and generate $M = 20$ compressions for each of the problem contexts.

#### D.1.3    QASPER

QASPER is a non-synthetic QA benchmark consisting of 1,585 scientific research papers in Natural Language Processing and 5,049 human-written questions about the content of the paper. Each scientific paper has up to 16,000 tokens (Dasigi et al., 2021). The questions are written by an NLP practitioner prior to reading the full paper, so QA evidence can be dispersed across multiple parts of the document. We subsample $N = 20$ documents and queries and generate $M = 20$ compressions for each of the documents. All experiments are run with $S = 3$ random seeds.

#### D.1.4    WILDCHAT

Our motivation in constructing a chat memory dataset is to simulate real-world memory systems that require models to integrate information across multiple previous interactions. Queries could build upon multiple previous exchanges, or individual isolated chats. In the original WILDCHAT dataset consisting of 837,989 multi-turn ChatGPT chats, each chat conversation exists as a standalone sample. We subsample $D = 1000$ chat conversations with between 4 and 8 turns to construct our dataset. The dataset construction process is as follows:

1. **User Construction:** We construct synthetic users by grouping 10 chat samples to each user (total $N = 100$ users).
2. **QA Generation:** We format each of the 10 chat conversations and provide GPT-4O-MINI with all full chat conversations along with the QA prompt to generate a question unique to each user that has not appeared in its chat history.

---

**QA Prompt Template**

```
You are a data generation assistant, tasked with building a benchmark that evaluates the memory
capabilities of a language model.
You will be provided a list of previous chat conversations.  Your goal is to generate a new synthetic
query that has not appeared in previous chats, but nevertheless benefits from the information in previous
chats.

CHATS:
{chats}

Generate a new synthetic query that has not appeared in previous chats, but nevertheless benefits from
the information that has appeared in previous chats.
Do not generate a RAG query about existing data in the chats, but rather a new query that could leverage
existing chat information as **memory**.
```

---

```
Please respond in the following JSON format:  <briefly think about the information you have and the
question you can generate from it>

{{
    "question":  "<question>",
    "answer":  "<answer>"
}}
Your answer (YOU MUST ONLY RESPOND WITH THE JSON OBJECT):
```

### D.1.5 FINEWEB

The FINEWEB dataset contains an extensive set of web pages since 2013. At the time of writing, the dataset includes 25.9 billion entries spanning from 2013 to 2025. To construct our subset of document and QA pairings, we collect $N = 100$ samples with between 15,000 and 28,000 tokens, and ask GPT-4O-MINI to synthetically generate 2 extractive and 3 creative QAs based on the cleaned web data and QA prompt.

---

**QA Prompt Template**

```
You are generating synthetic question-answer (QA) pairs from a source text.

SOURCE_TEXT:
{context}

Use only information from SOURCE_TEXT. No hallucinated facts.
Generate five questions and answers:
- Question 1:  What is {{topic}} and why is it important?  (type = "qa")
- Question 2:  What is {{topic}} and how does it work?  (type = "qa")
- Question 3:  Write an email to a colleague summarizing the findings and take-aways.  (type =
"generation")
- Question 4:  Generate rap lyrics that teach the core concepts.  (type = "generation")
- Question 5:  Generate a poem about the topic.  (type = "generation")

Please respond in the following JSON format:  <briefly think about the information you have and questions
you can generate from it>

{{
    "questions":  [
        {{
            "topic":  "<topic 1>",
            "question":  "<question 1>",
            "answer":  "<answer 1>",
            "type":  "qa"
        }},
        {{
            "topic":  "<topic 2>",
            "question":  "<question 2>",
            "answer":  "<answer 2>",
            "type":  "qa"
        }},
        {{
            "topic":  "<topic 3>",
            "question":  "<question 3>",
            "answer":  "<answer 3>",
            "type":  "generation"
        }},
        {{
            "topic":  "<topic 4>",
            "question":  "<question 4>",
            "answer":  "<answer 4>",
            "type":  "generation"
        }},
        {{
            "topic":  "<topic 5>",
            "question":  "<question 5>",
            "answer":  "<answer 5>",
            "type":  "generation"
        }}
    ]
}}

Your answer (YOU MUST ONLY RESPOND WITH THE JSON OBJECT):
```

---

### D.2 COMPRESSOR MODEL DETAILS

For the LLAMA-3 family, we use the models LLAMA-3.2-1B-INSTRUCT, LLAMA-3.2-3B-INSTRUCT, LLAMA-3.1-8B-INSTRUCT. For the QWEN-2.5 family, we use the models QWEN-2.5-1.5B-INSTRUCT, QWEN-2.5-3B-INSTRUCT, QWEN-2.5-7B-INSTRUCT. For the GEMMA-3 family, we use the models GEMMA-3-1B-IT, GEMMA-3-4B-IT, and GEMMA-3-12B-IT. Additionally, we evaluate QWEN-2.5-14B-INSTRUCT as compressor LM on WILDCHAT and FINEWEB.

For reasoning compression models, we use the models QWEN-3-4B and QWEN-3-8B, and for mixture-of-experts models, we use QWEN-3-30B-A3B.

All compressor model families are fine-tuned for instruction following. Compression outputs of at most 4096 tokens are generated with temperature of 0.7 for LLAMA-3, QWEN-2.5, and QWEN-3, and 1.0 for GEMMA-3.

## D.3 PREDICTOR MODEL DETAILS

As predictor models we use GPT-4O, LLAMA-3.1-8B-INSTRUCT, LLAMA-3.3-70B-INSTRUCT, and LLAMA-3.1-405B-INSTRUCT. Predictor models generate with a temperature of 0.6 across all benchmarks and experiments.

## D.4 GENERALIZED LINEAR MODEL ANALYSIS SETUP

We fit a logistic regression that predicts binary correctness of a compression-prediction output on:

- Z-score normalized lengths of the input document, prediction output, and compression output,
- Z-score normalized predictor and compressor model size,
- Indicator $\mathbf{1}\{\text{Compressor=Qwen}\}$ for the compressor model family,

where the predictors are LLAMA-3 models of sizes 1B, 8B, 70B, and 405B.

## D.5 MUTUAL INFORMATION PROXY MODEL DETAILS

We choose a QWEN-2.5-7B proxy model when evaluating LLAMA compressors and a LLAMA-3.1-8B proxy model when evaluating QWEN-2.5 and GEMMA-3 compressors. We directly use internal log probabilities to estimate mutual information for QWEN-3 compressors.

## D.6 DEEP RESEARCH SETUP DETAILS

For our experiments, we randomly sample $N = 20$ English research tasks from the DEEPRESEARCH BENCH test set to ensure a representative evaluation across diverse research domains. We conduct 5 independent runs for each experimental configuration. This allows us to report mean performance with standard error bars, providing a robust assessment.

### D.6.1 FULL DEEP RESEARCH WORKFLOW SETUP

In our Deep Research system setting, a predictor LM decomposes each research task into a collection of *(Query, Subtask)* pairs. Each pair consists of a targeted web search query with a natural language instruction that specifies how the retrieved evidence should be analyzed. The predictor then distributes these pairs to compressor LMs, which independently perform the searches in parallel. Compressor LMs process the retrieved content according to the subtask, and compress the results into summaries. The predictor then aggregates these summaries into a comprehensive research report. This setup is illustrated in Figure 8.

We evaluate our system using the DEEPRESEARCH BENCH framework (Du et al., 2025), which assesses agent performance through four dimensions: Comprehensiveness, Depth, Instruction-following, and Readability. More specifically, we use *RACE* (Reference-based Adaptive Criteria Evaluation) scores to study the impact of model scale. The costs used in Figure 7 are based on GPT-4O API rates (Aug 2025: $2.50/1M input tokens, $10.00/1M output tokens). In addition, there is a constant SerpAPI web search cost of $0.12 for every task, which is not included in the figure.

### D.6.2 DEEP RESEARCH COMPRESSOR MODEL DETAILS

We employ the QWEN-2.5-INSTRUCT family of models as compressor LMs, ranging from 0.5B to 14B parameters. These models are hosted on Modal Labs using the SGLang inference framework, enabling free, high-throughput parallel inference. All compressor models use a temperature of 0.7 and a maximum output token limit of 2,000 tokens per response. The specific compressor models used are:

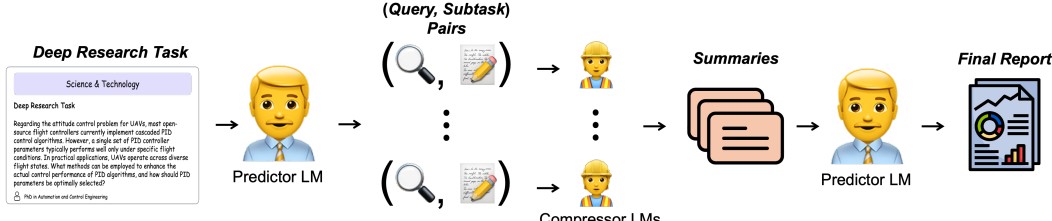

Figure 8: **Deep Research workflow.** A predictor LM decomposes a Deep Research task into *(Query, Subtask)* pairs, where each pair specifies a targeted web search and an associated analysis instruction. Compressor LMs work in parallel to retrieve evidence, process it according to the subtask, and compress the findings into concise summaries, which the predictor then aggregates into a final report.

- QWEN-2.5-0.5B-INSTRUCT: Smallest model for minimal compression overhead.
- QWEN-2.5-1.5B-INSTRUCT: Balance between efficiency and capability.
- QWEN-2.5-3B-INSTRUCT: Mid-range compression quality.
- QWEN-2.5-7B-INSTRUCT: Strong comprehension with moderate compute.
- QWEN-2.5-14B-INSTRUCT: Highest quality compression in our experiments.

Each compressor independently processes the search results for its assigned *(Query, Subtask)* pair, extracting and compressing the relevant information according to the predictor's instructions. The compressed summaries from all compressors are then aggregated by the predictor into the final research report.

### D.6.3    DEEP RESEARCH PREDICTOR MODEL DETAILS

We evaluate four predictor models spanning different scales and providers:

- LLAMA-3.1-8B-INSTRUCT: Entry-level predictor with basic task decomposition capabilities. Temperature set to 0.6, maximum output tokens of 4,000.
- LLAMA-3.1-70B-INSTRUCT: Mid-tier predictor with improved reasoning and task planning. Temperature set to 0.6, maximum output tokens of 4,000.
- LLAMA-3.1-405B-INSTRUCT: Large-scale predictor with advanced multi-step reasoning capabilities. Temperature set to 0.6, maximum output tokens of 4,000.
- GPT-4O: State-of-the-art commercial predictor serving as our performance upper bound. Temperature set to 0.6, maximum output tokens of 16,000 to accommodate comprehensive report generation.

All predictors use a slightly lower temperature (0.6) compared to compressors to ensure more consistent and structured task decomposition and report synthesis. The predictor is responsible for: (1) decomposing the research question into targeted queries, (2) formulating specific subtasks for each query, (3) distributing work to the compressor pool, and (4) synthesizing compressor outputs into a coherent final report.

A notable limitation is that the Llama family predictors are constrained to 4,000 output tokens, which can limit the comprehensiveness of their final research reports compared to GPT-4o's 16,000 token capacity. This constraint particularly affects the synthesis phase where the predictor must compile information from multiple compressor summaries into a cohesive report.

### D.7    COMPRESSION FAILURE MODES

We attempt to understand how scaling compressor size qualitatively increases downstream QA accuracy. To do so, we categorize compressor errors into one of the three recurring failure modes.

**Category 1: Incorrect answer.** The compression contains an erroneous number or finding in an attempt to answer the question, often due to confusion between reports at different time points. Larger compressors reduce this due to their capacity to better model temporal dependencies.

---

**Error Category 1: Incorrect Answer**

**Query:** When was Mr. Alan Fisher first diagnosed with a heart disease?
**Summary:** Mr. Alan Fisher was first diagnosed with coronary artery disease on 10/1995, during a routine renal transplant biopsy due to progressive deterioration of kidney function. The initial biopsy showed isometric tubular epithelial vacuolization, which was associated with isometric microvesicular cytoplasmic transformation, exceeding 80%. This finding was highly suggestive of Chronic Allograft Nephropathy (CAN).
**Prediction:** 10/1995
**Correct Answer:** 1989
**Evidence in context:** **Other Diagnoses** [...] Coronary artery disease, CAD-3: - Previous anterior wall infarction in 1989, treated with [...]
**Error diagnosis:** Other diagnoses were made in 10/1995, but not coronary artery disease.

---

**Category 2: No answer.** The compression contains no information relevant to the query. All facts and findings in the compression are unrelated, leaving the question unaddressed. The compressor model fails to include any answer-bearing content.

---

**Error Category 2: No answer**

**Query:** What significant cardiac intervention did Mr. Fisher undergo in May 1995?
**Summary:** Mr. Alan Fisher was admitted on March 14, 2020, and underwent a urethrotomy interna and calculated intravenous antibiotic therapy with Meropenem. He was discharged with a urinary catheter inserted. However, he still experienced recurrent urinary tract infections. An additional treatment plan was developed to include antifungal therapy with Fluconazole. The patient was discharged on March 19, 2020.
**Prediction:** None
**Correct Answer:** PTCA + stent in RIVA
**Evidence in context:** **Other Diagnoses** [...] PTCA + stent in the right coronary artery (RIVA) in 05/1995
**Error diagnosis:** The cardiac intervention in May 1995 was embedded as background information in the section "Other Diagnoses" and not directly in the main medical findings. The model failed to understand the importance of historical interventions to answering the query.

---

**Category 3: Details or parts of the findings are omitted from compression.** The compression may correctly identify all medical findings or records required to answer the query, however not in enough detail to adequately answer the question. The compression may also capture only part of the medical findings required to answer the query, while omitting certain facts/information that are crucial to answering the full question. We often observe this when the compression contains many different patient/treatment facts that are irrelevant for answering the query, such that the critical findings are glossed over and "overly compressed".

---

**Error Category 3: Omission of information/over-compression**

**Query:** Where did Mrs. Done show metastases?
**Summary:** Mrs. Jane Done, born on 01.01.1966, had a history of metastatic malignant melanoma, with the most recent metastases being detected on 11/01/2020. Her treatment included a microsurgical resection of a right frontal tumor in 2018, a 2020 radiation therapy course, and immunotherapy with Nivolumab and Ipilimumab. In 2021, she was diagnosed with cervical lymph node metastasis, and underwent surgery followed by adjuvant therapy with pembrolizumab. In 2022, she presented with multiple metastases in her CT scan, which were confirmed on PET-CT. Her most recent treatments include six cycles of Vemurafenib, Cobimetinib, and Pembrolizumab.
**Prediction:** Right frontal tumor, cervical lymph nodes, and multiple unspecified locations
**Correct Answer:** Brain, lungs, liver, heart, lymph nodes, muscles, bone
**Evidence in context:** Microsurgical resection right frontal tumor [...] hemorrhaged right frontal metastasis from previously diagnosed malignant melanoma [...] multiple roundish subsolid nodules found bipulmonary [...] multiple hypodense lesions throughout both lobes, indicative of metastatic spread [...] concerning 2 cm mass abutting the lateral wall of the left ventricle raising the suspicion for cardiac metastasis [...] Cervical lymph node metastasis [...] a 2.5 cm mass identified within the left psoas muscle, consistent with muscular metastasis [...] lytic lesions involving the sternum and right 4th rib, consistent with osseous metastatic disease
**Error diagnosis:** The compressor selectively included only frequent metastasis mentions explicitly (brain, lymph nodes) in its summary while compressing numerous organ-specific findings in other parts of the context (lungs, liver, heart, muscles, bone) as "multiple metastasis".
This suggests that the compressor model was successful in identifying further metastasis. However, the compressor model did not provide all details necessary for answer completeness and was overly aggressive in compressing sites mentioned less frequently in the context.

---

# E    EXTENDED RESULTS

In this section, we present extended results and ablations of key design choices in our compression-prediction setup.

## E.1    EXTENDED RESULTS ON SCALING LAWS OF COMPRESSOR MODELS

We extend our analysis by constructing synthetic QA tasks on three further datasets (two synthetic, one non-synthetic) and evaluate the accuracy and perplexity of the compressions across different compressor model sizes. We measure perplexity by evaluating the log probabilities of a LLAMA-3.1-8B model on the target answer.

### E.1.1 SUMMARIZING SCIENTIFIC PAPERS ON QASPER

On QASPER, we extend our compressor scaling analysis to compression-prediction workflows on non-synthetic scientific papers. We vary compressor size for non-reasoning LLAMA and QWEN-2.5 and reasoning QWEN-3 compressors.

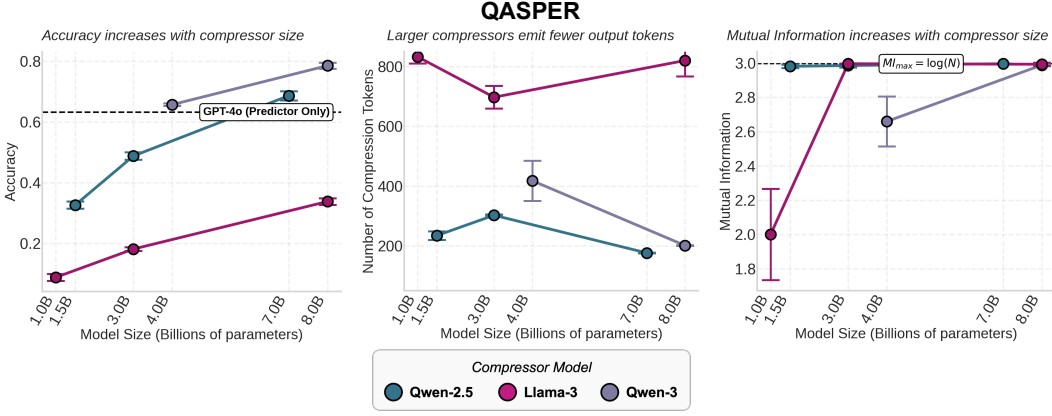

Figure 9: **Scaling behavior holds for reasoning and mixture-of-experts compressor models on QASPER.** We scale compressor model size and reports a different metric of the compression step on the $y$-axis of each column: **(Left)** accuracy, **(Middle)** compression length, **(Right)** mutual information. Mutual information is estimated using the log probabilities of a proxy model for QWEN-2.5 and LLAMA compressors, and using internal log probabilities for QWEN-3 compressors. Larger compressors produce shorter outputs with higher downstream accuracy and higher mutual information.

**Larger compressors are more accurate.** We find that compressions generated by larger compressor models are more accurate across all three model families. Compressors at the 8B scale outperform the GPT-4O-only baseline (Figure 9).

**Larger compressors retain more mutual information.** Scaling trends observed on LONGHEALTH and FINANCEBENCH continue to hold on the QASPER dataset. Larger compressors output summaries of the scientific papers that carry more mutual information, with models at the 8B scale yielding up to $1.5\times$ more mutual information.

### E.1.2 CONSTRUCTING CHAT MEMORY ON WILDCHAT

In our experiments, a compressor model summarizes long contexts with regard to context-specific questions. In practice, long context lengths also pose a major challenge in recalling information from past LM chat conversations (Eyuboglu et al., 2025). Modern LLM chatbots construct internal memory about a user's past chat histories, which serve as context for future conversations. Instead of generating query-specific summaries, we generate chat memories for each user by summarizing each chat interaction of a user using a compressor LM. The predictor then attempts to answer synthetic queries posed by the user based on the chat memory. Again, we vary the compressor model size and examine its effects on downstream perplexity, compression size, and compute cost in FLOPs-per-generation.

**Larger compressors yield lower perplexity.** As expected, we find in Figure 10 that chat memories generated by larger compressor models yield lower perplexity across model families. Query-agnostic summaries of chat conversations output by the largest compressor model of each model family yield up to $1.14\times$ lower log probabilities as compared to the 1B model sizes.

**FLOPs-per-generation scaling holds on WILDCHAT.** The scaling of compression output length on holds for QWEN-2.5 and GEMMA-3 compressors, resulting in FLOPs-per-generation scaling

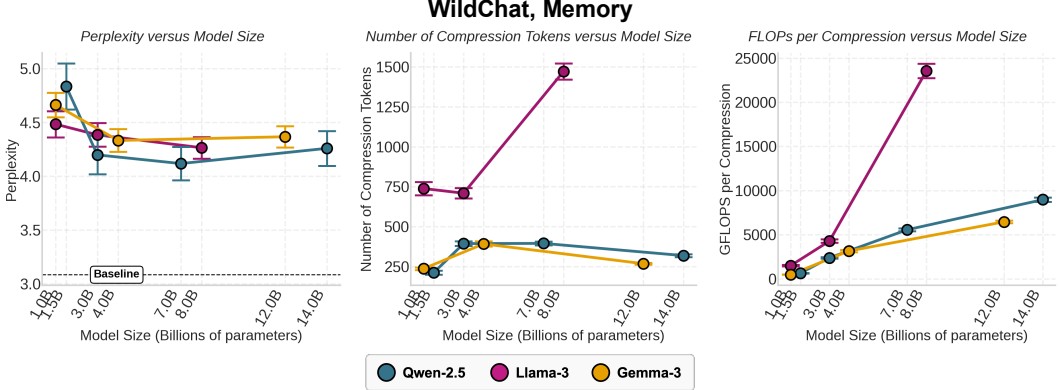

Figure 10: **Perplexity, compression length, and compute cost scale with compressor size (WILDCHAT).** We scale compressor model size and report a different metric on the $y$-axis of each column: **(Left)** Perplexity, with the black dashed line showing baseline perplexity given all 10 full chat conversations, **(Middle)** compression length of each set of chat conversations, **(Right)** GFLOPs-per-compression. Larger compressors produce shorter outputs with lower perplexity.

sublinearly with model size (Figure 10). In contrast, larger LLAMA compressors generate longer compressions, resulting in steeper scaling of FLOPs-per-generation. However, we observe consistent trends in scaling of FLOPs-per-generation between compressor model families: It is significantly more compute-efficient to scale QWEN-2.5 and GEMMA-3 compressors than LLAMA compressors.

### E.1.3 VARYING TASK TYPE ON FINEWEB

We extend our analysis of scaling compressor model size to a fourth dataset. On FINEWEB, we further ablate by task type: **extractive** tasks, which require the predictor model to identify and reproduce information explicit in the context—e.g. factual QA—and **creative** tasks which require the predictor model to generate longer, open-ended outputs that is not verbatim in the context—e.g., paraphrasing, format-change). We examine the compressor scaling behavior of both query-specific (Figure 11) and query-agnostic (Figure 12) summaries.

**Larger compressors yield lower perplexity.** As expected, we find that increasing compressor size consistently reduces perplexity for both extractive and creative tasks, on both query-specific and query-agnostic summaries. Larger compressors approach the baseline performance of giving the predictor direct access to the full uncompressed context, rather than a lossy compression thereof (Figures 11, 12).

We observe that perplexity differs in magnitude for different task types. Extractive tasks show lower perplexity values, as answers are explicitly present in the context, while creative tasks are more challenging. Query-agnostic summaries tend to achieve lower perplexity on creative tasks than query-specific summaries, which suggests that broader, more general compressions capture stylistic and semantic cues that are key to creative, open-ended generation tasks.

**FLOPs-per-generation scaling holds on FINEWEB.** We continue to observe predictable scaling of compute cost. As we increase compressor size, the amount of FLOPs-per-generation increases at different rates consistent with our findings on LONGHEALTH, FINANCEBENCH, and WILDCHAT. Scaling QWEN-2.5 and GEMMA-3 compressor model size comes at a cheaper compute cost than for LLAMA compressors. We find identical trends across different natures of the task (extractive vs. creative) and types of summary (query-specific vs. query-agnostic).

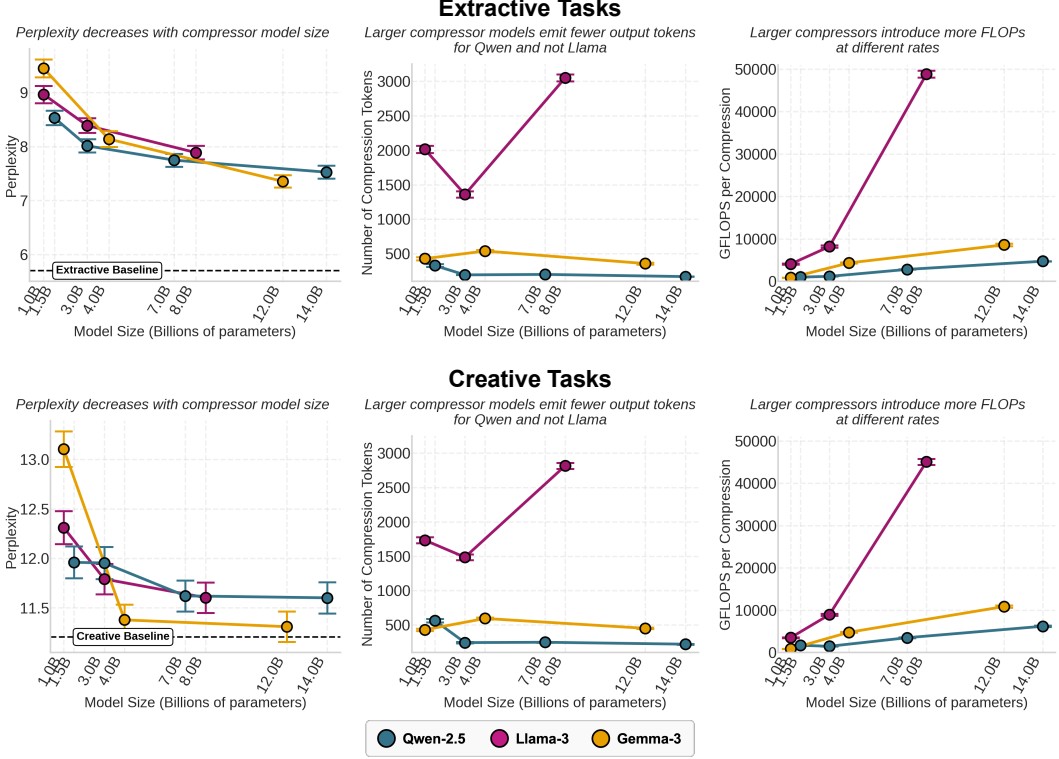

Figure 11: **Perplexity, compression length, and compute cost scale with compressor size (FINEWEB; Top: Extractive Tasks; Bottom: Creative Tasks).** We scale compressor model size and report a different metric on the $y$-axis of each column: **(Left)** perplexity, with the black dashed line showing baseline perplexity given the full context, **(Middle)** compression length, **(Right)** GFLOPs-per-compression. Larger compressors produce shorter outputs with lower perplexity.

### E.1.4 EFFECT OF PROXY MODEL ON MUTUAL INFORMATION ESTIMATION

As discussed in Section 2.2, proxy models are used to evaluate the log probabilities of compressions generated by small LMs when estimating mutual information. This is necessary when the compressor is insufficiently calibrated.

To understand the effect of the choice of proxy model in our mutual information estimator, we compare three distinct proxy LMs at the 7–8B scale: QWEN-2.5-7B, QWEN-3-8B, and LLAMA-3.1-8B (Figure 13). We find that the choice of proxy model introduces a fixed vertical offset in the MI curves that is consistent across estimates on LONGHEALTH. However, it does not affect the scaling rates or any of the previous conclusions drawn.

### E.1.5 EXTENDED RESULTS ON REASONING AND MIXTURE-OF-EXPERTS COMPRESSORS

We present initial results for reasoning and mixture-of-experts compressor models. Specifically, we compare how dense non-reasoning (QWEN-2.5), dense reasoning (QWEN-3), and MoE reasoning (QWEN-3-30B-A3B) compressors scale across accuracy, compression length, and mutual information. We observe similar compressor scaling trends across reasoning and non-reasoning compressors. Interestingly, the mixture-of-experts model outperforms dense models of the same scale in accuracy, producing more concise compressions with higher mutual information (Figure 14).

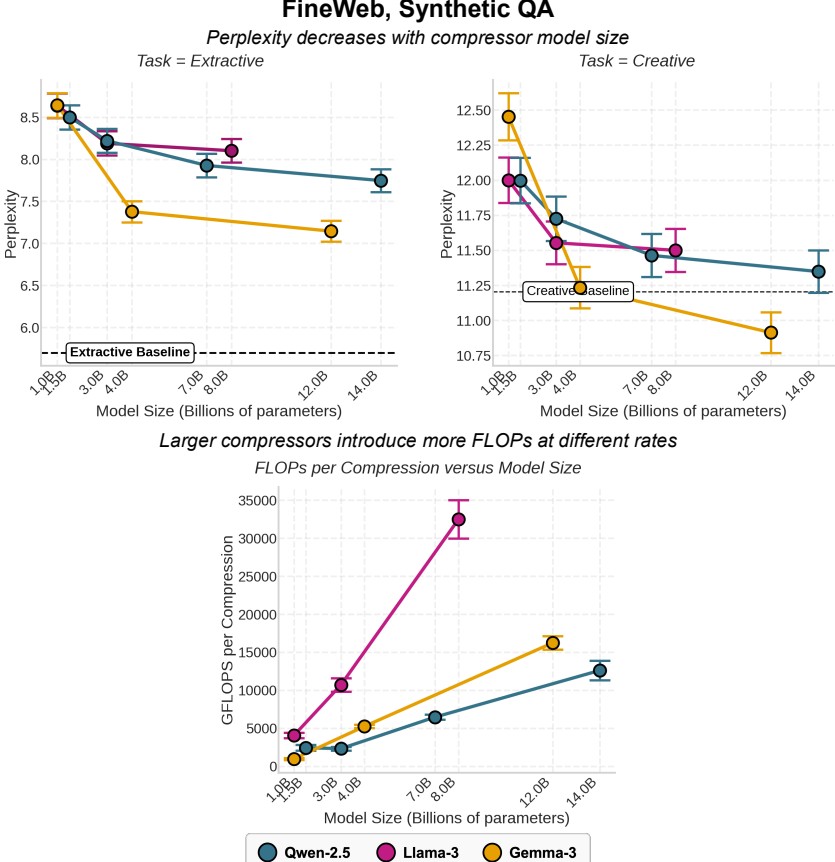

Figure 12: **Perplexity, compression length, and compute cost scale with compressor size for query-agnostic compressions (FINEWEB).** We scale compressor model size on different types of tasks (extractive and creative) and report a different metric on the $y$-axis of each subplot: **(Top Left)** perplexity evaluated on extractive tasks, **(Top Right)** perplexity evaluated on creative tasks, **(Bottom)** compression length. Larger compressors produce query-agnostic compressions with lower perplexity across both types of tasks.

### E.1.6  SCALING OF MUTUAL INFORMATION AND BIT EFFICIENCY ON FINANCEBENCH

To understand whether our information-theoretic findings generalize beyond LONGHEALTH, we examine mutual information and bit efficiency scaling on FINANCEBENCH. Figure 15 shows that the scaling behavior remains consistent: larger compressors retain more information about the original document while compressing more efficiently.

### E.1.7  WHAT ARE THE EFFECTS OF CONCISENESS INSTRUCTIONS?

We ask whether explicitly instructing the compressor to different levels of conciseness changes the scaling behaviors that we observe on FINANCEBENCH. We vary the prompt to instruct the compressor LM to be *concise* (3 sentences), *normal* (6 sentences), and *elaborate* (9 sentences). We find in 16 that accuracy and MI are unaffected by instructed conciseness on FINANCEBENCH. While prompting shifts compression output size and compute cost by an absolute offset, the compressor scaling trends hold across different conciseness constraints, showing that our scaling results are driven by compressor capacity.

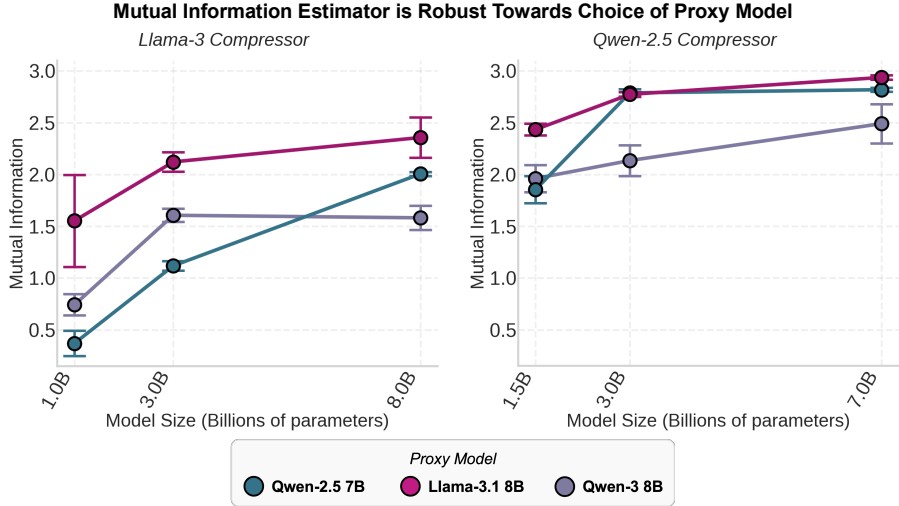

Figure 13: **Monte Carlo mutual information estimator is robust towards choice of proxy model.** The $y$-axis shows the mutual information estimate on LONGHEALTH compressions produced by **(Left)** LLAMA and **(Right)** QWEN-2.5 compressor models. The choice of proxy model introduces a fixed vertical offset in MI estimate, but does not affect compressor scaling behavior.

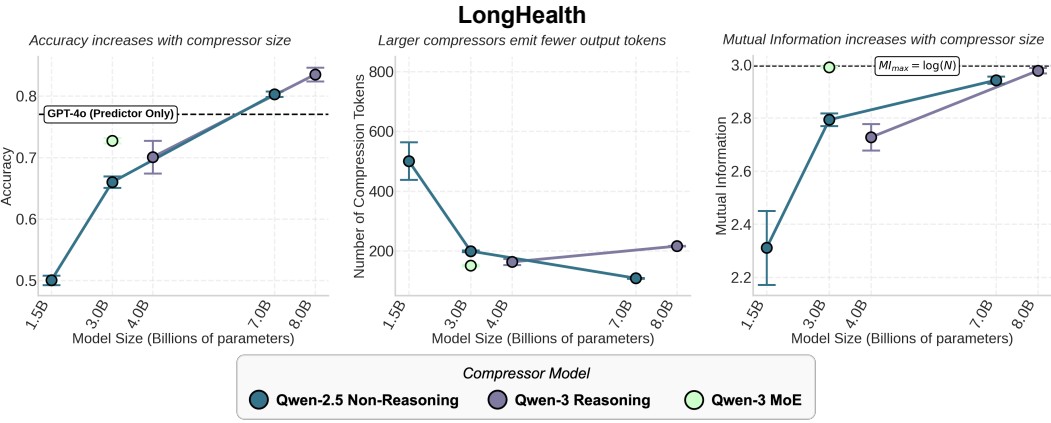

Figure 14: **Compressor scaling behavior holds for reasoning and mixture-of-experts models.** We scale compressor model size and reports a different metric of the compression step on the $y$-axis of each column: **(Left)** accuracy, **(Middle)** compression length, **(Right)** mutual information. Mutual information is estimated using the log probabilities of a proxy model for non-reasoning QWEN-2.5 compressors and using internal log probabilities for dense and MoE reasoning QWEN-3 compressors. Scaling trends are consistent with our observations for non-reasoning dense models (blue), where larger compressors yield higher accuracy with shorter compressions and higher mutual information. At the same scale (3B), the mixture-of-experts model (green) outperforms the dense models in downstream accuracy, compression conciseness, and mutual information.

### E.1.8 MULTI-TURN INTERACTIONS

We extend our analysis beyond the single-turn compression-prediction setting to evaluate multi-turn workflows on LONGHEALTH.

In our setup, the predictor (LLAMA-3.1-405B) is allowed to query the compressor (LLAMA-3.2-3B) for additional information for three rounds. At each turn, the predictor integrates the compression it

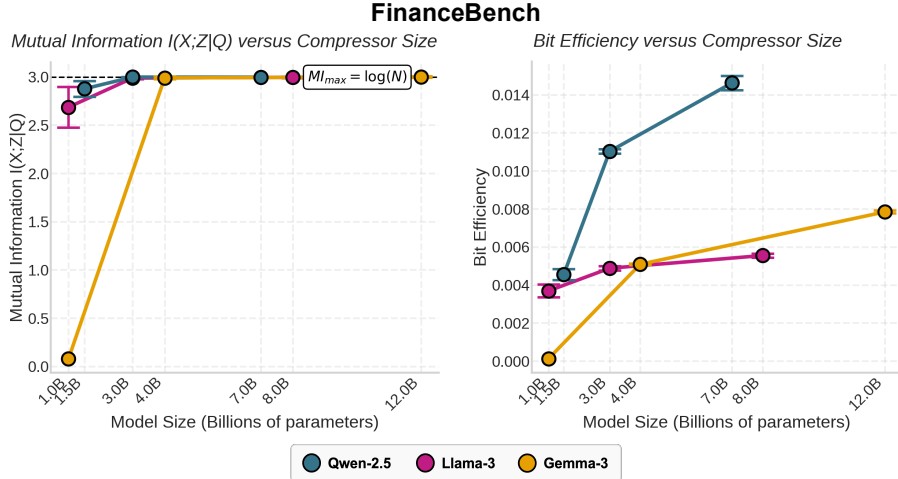

Figure 15: **Larger compressors generate outputs that carry more information about their inputs (conditioned on the query) on FINANCEBENCH.** We scale compressor model size and estimate the **(Left)** mutual information, and **(Right)** bit efficiency (bits of mutual information per token; higher is better) carried by their outputs. Larger compressor model sizes compress documents with higher mutual information and bit efficiency.

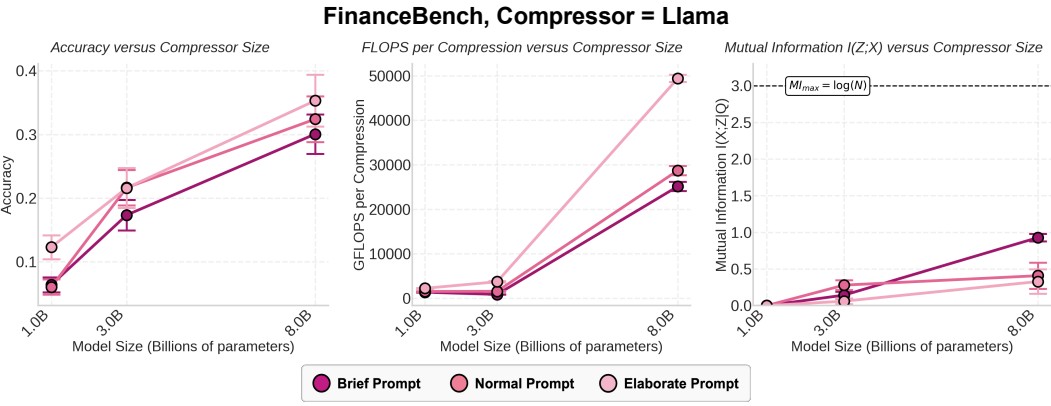

Figure 16: **Scaling behavior of compressor model size hold across instructed conciseness (COMPRESSOR = LLAMA).** We ablate over compression conciseness levels by varying the compression prompt instructions. We measure **(Left)** accuracy, **(Middle)** GFLOPs-per-generation, and estimate **(Right)** mutual information. We find that accuracy and mutual information are largely unaffected by conciseness instructions. Compressors instructed to be more concise are more token-efficient, and thus compute-efficient. Trends in accuracy, compute cost, and mutual information as we scale compressor hold across conciseness constraints.

has constructed so far with the information it has received this turn, and issues a targeted follow-up query asking for the most relevant information, effectively separating data and control plane.

We show in Figure 18 that the amount of mutual information the compressions carry at each turn increases when given the opportunity to query more information. However, additional turns past two rounds do not provide further improvements.

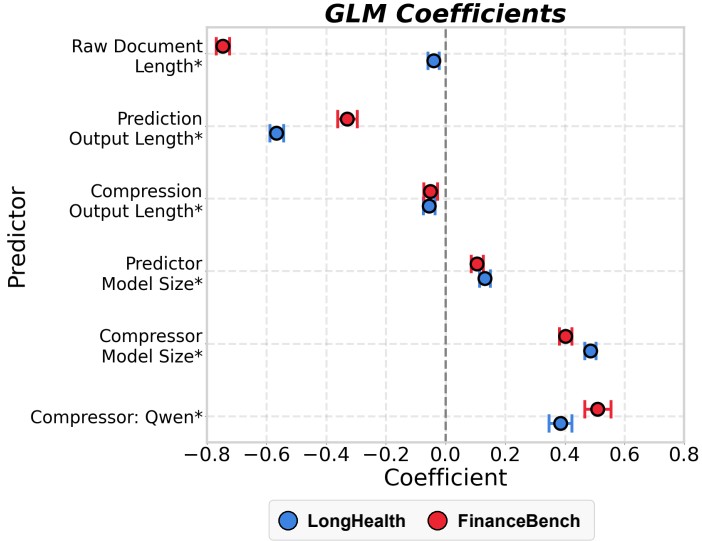

Figure 17: **Generalized Linear Model (GLM) Coefficients.** We conduct regression analysis on a GLM predicting QA correctness (0/1) on **(Blue)** LONGHEALTH, **(Red)** FINANCEBENCH. The y-axis shows coefficient estimates for each variable, horizontal bars are 95% confidence intervals, asterisks mark variables that are significant at $p < 0.05$ on both datasets. For more details on our GLM setup, refer to Appendix D.4.

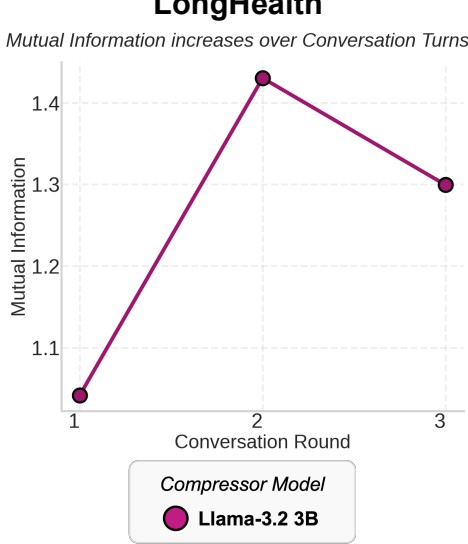

Figure 18: **Mutual information in multi-turn workflows on LONGHEALTH.** We allow a LLAMA-3.1-405B predictor to repeatedly query a LLAMA-3.2-3B compressor for additional information. Increasing the compression-prediction workflow to two turns yields an increase in mutual information, but we observe no additional improvement when extending to a third turn.

### E.2    EXTENDED RESULTS OF RATE-DISTORTION ANALYSIS

We aim to establish rules of thumb for design decisions around choice of compressor and predictor models based on rate-distortion theoretic concepts introduced in Section 2.2 and Appendix B.3. We further investigate the fidelity, compute cost, and communication efficiency of different compressor-

predictor pairings. We examine QWEN-2.5 and LLAMA compressor models from 1B to 8B, and QWEN-2.5 and LLAMA predictor models stretching from 1B to 405B parameters.

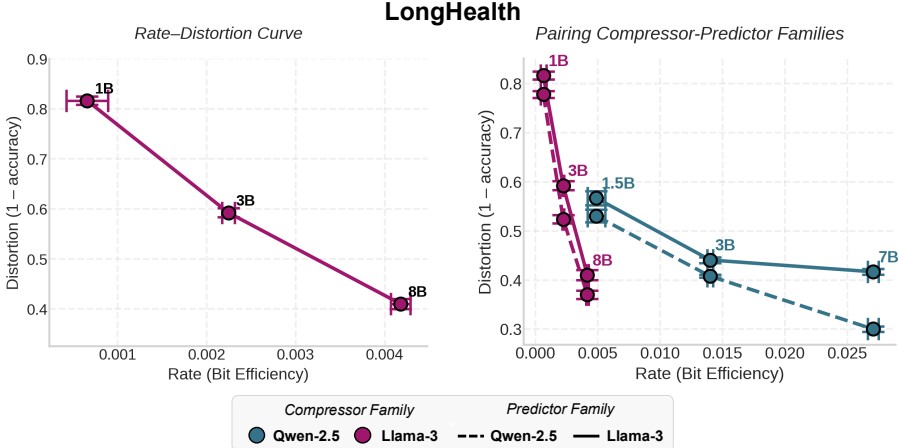

Figure 19: **Exploring the trade-off between compression and fidelity loss: rate-distortion curve.** We vary predictor and compressor model in the compression-prediction workflow and measure the distortion on the $y$-axis and estimate the rate on the $x$-axis. **(Left)** We examine a single compressor-predictor LM pairing, COMPRESSOR=LLAMA-3 and PREDICTOR=LLAMA-3.3-70B. **(Right)** We compare different compressor-predictor LM pairings, where the predictor model is QWEN-2.5-72B ("Qwen-2.5") or LLAMA-3.3-70B ("Llama"). Markers indicate compressor sizes (1B, 3B, 8B) in the LLAMA-3 compressor model family.

### E.2.1 SCALING PREDICTOR MODEL SIZE

In a compression-prediction system, the compressor model acts as a bottleneck on information about the document $X$. If we fix that information bottleneck and the amount of information passed through, how does the capacity to generate predictions affect downstream QA performance? We fix the compressor model to be either QWEN-2.5 or LLAMA, and vary the predictor to be LLAMA models of sizes 1B, 8B, 70B, 405B on the datasets LONGHEALTH and FINANCEBENCH (Figure 20). By plotting accuracy against bit efficiency in an alternative representation of rate-distortion analysis, downstream QA accuracy improves when moving from small predictors at 1B to larger 8–70B predictors, but then saturates with further scaling to a massive 405B predictor (Figure 21).

### E.2.2 ALTERNATIVE MEASUREMENTS OF DISTORTION

In addition to our primary measurement of distortion as 1–accuracy, we evaluate an alternative notion of distortion based on semantic similarity. We embed both prediction and target answer using the OpenAI TEXT-EMBEDDING-3-SMALL embedding model, and measure distortion as 1–cosine similarity between the semantic embeddings.

We observe characteristic rate-distortion curves with the same ordering of predictor model sizes (Figure 23).

### E.3 EXTENDED RESULTS OF DEEP RESEARCH ANALYSIS

In our Deep Research scaling experiments, we additionally ablate the effect of search result quantity by providing GPT-4O with the top $k = 1, 3, 6$ search results from each of the 8 predictor queries directly, bypassing the compression step. This analysis in Figure 7 reveals that even with maximal context utilization ($k = 6$, totaling 48 sources), the uncompressed approach achieves a similar level of *RACE* scores at significantly higher API costs compared to our compression-based strategy.

We also compute the compute cost in FLOPs-per-generation of the system, as shown in Figure 24. We observe an increase in the number of total FLOPs-per-generation used by the Deep Research

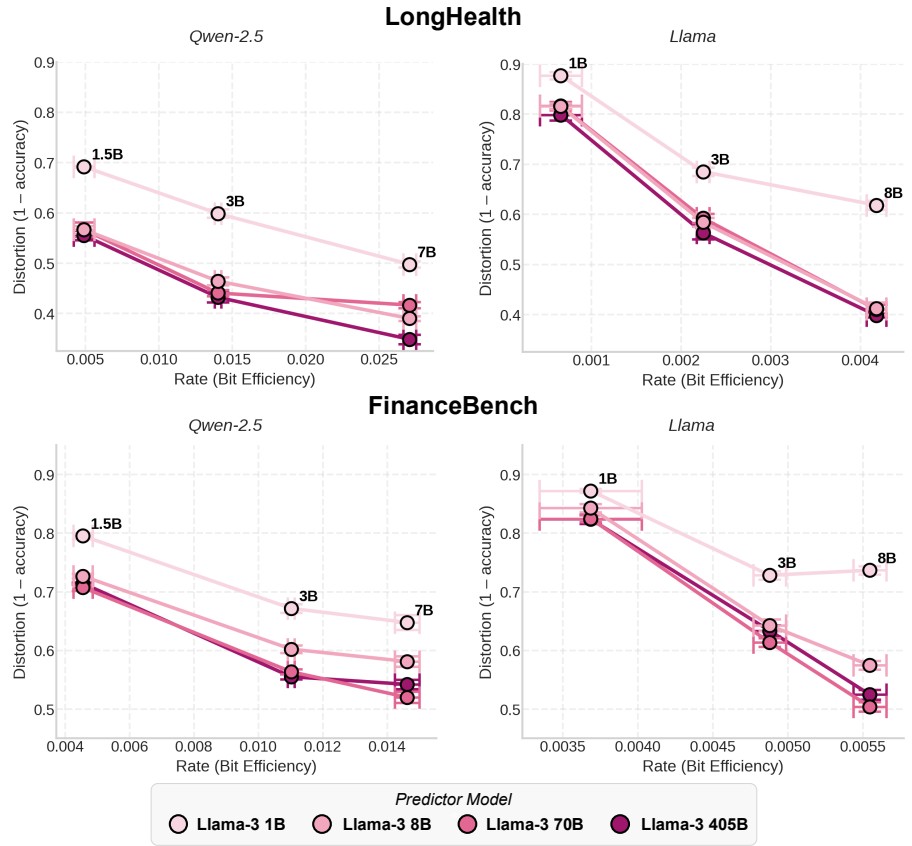

Figure 20: **Exploring the trade-off between compression and fidelity loss: rate-distortion curve.** We vary both predictor and compressor model in the compression-prediction workflow and measure the distortion on the $y$-axis and estimate the rate on the $x$-axis. We plot the resulting rate-distortion curves across predictor sizes 1B, 8B, 70B, and 405B for **(Left)** QWEN-2.5 and **(Right)** LLAMA compressors. We evaluate the rate-distortion curves for two datasets: **(Top)** LONGHEALTH, **(Bottom)** FINANCEBENCH.

system as predictor size increases. We generally observe that larger compressors extract and provide more tokens of information in their compressions.

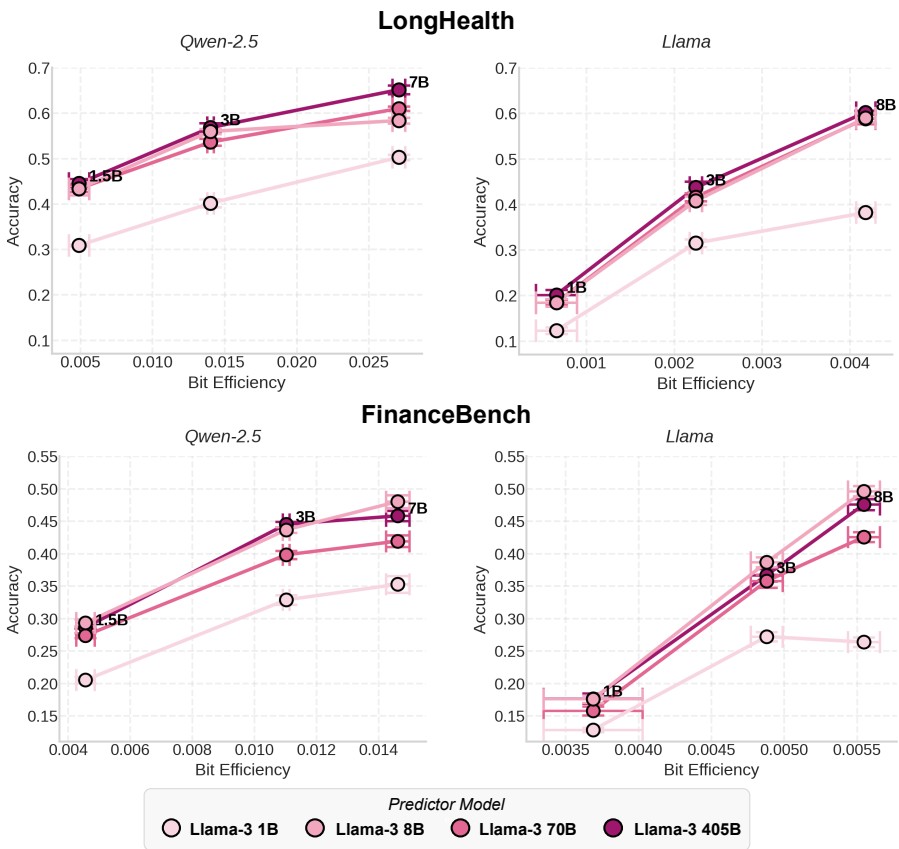

Figure 21: **Exploring the relationship between compression and accuracy.** The y-axis depicts accuracy and the x-axis shows bit efficiency. *Bit efficiency* is defined as the bits of mutual information encoded in each compression token. Markers indicate compressor sizes in the QWEN-2.5 (1.5B, 3B, 7B) and LLAMA (1B, 3B, 8B) compressor model family; vertical and horizontal bars denote standard errors.

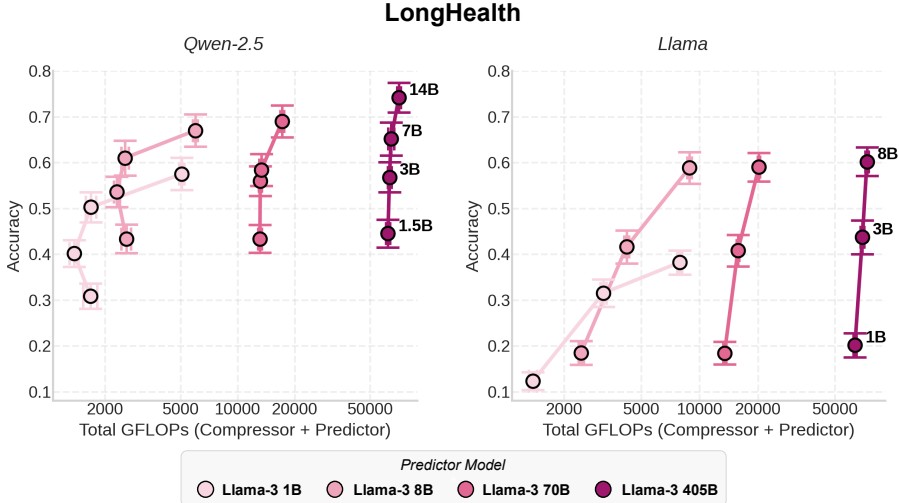

Figure 22: **QA Accuracy versus total compute cost on FINANCEBENCH.** In each panel, the y-axis shows the accuracy and the x-axis plots total compute cost in FLOPs-per-generation on a log-scale for **(Left)** QWEN-2.5, **(Right)** LLAMA-3 compressor LMs. Markers indicate compressor sizes in the QWEN-2.5 (1.5B, 3B, 7B) and LLAMA-3 (1B, 3B, 8B) compressor model family; vertical and horizontal bars denote standard errors.

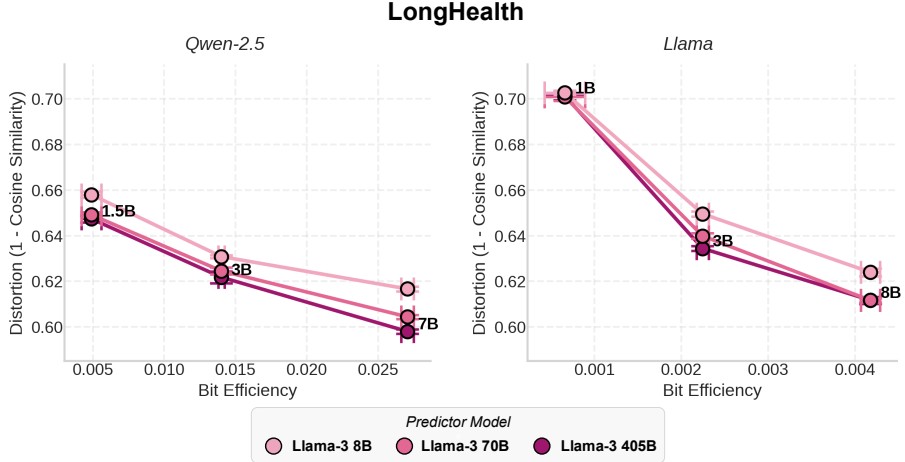

Figure 23: **Exploring the trade-off between compression and fidelity loss: alternative definition of distortion (LONGHEALTH).** We vary both predictor and compressor model in the compression-prediction workflow and measure the distortion on the $y$-axis as 1–cosine difference between the semantic embedding of the prediction and target answer. We estimate the rate on the $x$-axis. We plot the resulting rate-distortion curves across predictor sizes 8B, 70B, and 405B for **(Left)** QWEN-2.5 and **(Right)** LLAMA compressors.

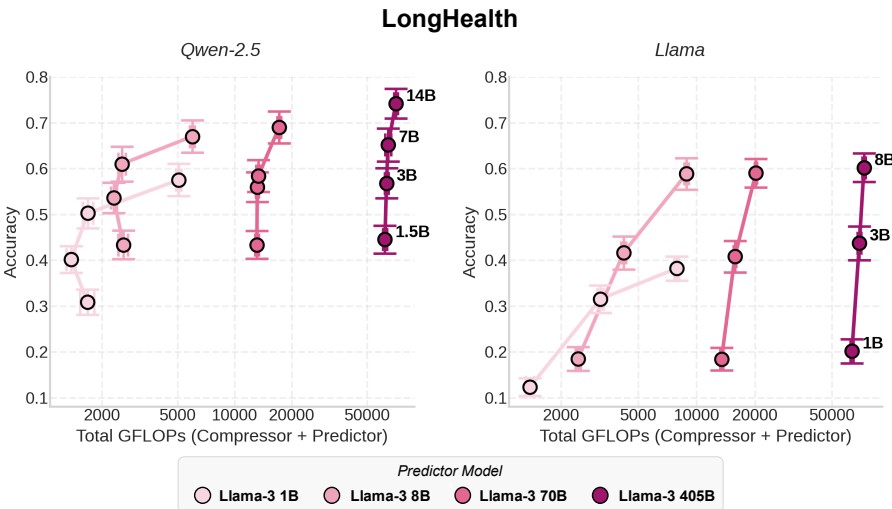

Figure 24: **Compressor model size versus compute and token usage.** In each panel, the x-axis shows the Qwen compressor size (in billions of parameters). **(Left)** Total GFLOPs per task grows with compressor model size, with larger predictors (Llama 405B, 70B, 8B) amplifying compute cost. **(Right)** Compressor output tokens per task, which remain relatively stable across predictors (GPT-4o, Llama 405B, 70B, 8B), increase moderately with larger compressors.

