# OpenReview forum: "An Information Theoretic Perspective on Agentic System Design"
_ICLR.cc/2026/Conference — ICLR 2026 Poster_

### Official Review · Reviewer_H8yM · 2025-10-15

**Soundness:** 3
**Presentation:** 3
**Contribution:** 3
**Rating:** 6
**Confidence:** 4

**Summary:**

This paper presents a novel, information-theoretic perspective on designing agentic language model systems, particularly those following a "compressor-predictor" architecture. The authors argue that the current design process for such systems is ad-hoc. To address this, they propose using Mutual Information (MI) between the source context and the compressed summary as a task-agnostic metric to evaluate the quality of the "compressor" LM. Through extensive experiments across four datasets, the paper derives several key design principles. The most significant finding is that scaling the compressor model is substantially more effective for improving downstream performance than scaling the predictor model. The authors also show that larger compressors are more token-efficient and that the proposed MI-based metric strongly correlates with final task performance, offering a valuable proxy for system evaluation.

**Strengths:**

* **Novel and Principled Framework**: The primary strength of this paper is its novel application of information theory to a very practical and important problem. Moving the design of agentic systems from ad-hoc trial-and-error to a more principled framework grounded in metrics like Mutual Information is a significant conceptual contribution.
* **Actionable and Impactful Findings**: The paper produces clear, actionable, and somewhat counter-intuitive design principles. The core finding—that it is more effective to "front-load" compute by investing in a more powerful compressor rather than a larger predictor—has significant practical implications for system design, especially regarding the trade-off between local (on-device) compute and expensive cloud API calls.
* **Comprehensive Empirical Analysis**: The authors conduct an extensive set of experiments across multiple model families, model sizes, and datasets to validate their claims. The scaling law analyses are thorough and provide strong evidence for their conclusions regarding the relative importance of the compressor versus the predictor.

**Weaknesses:**

* **Methodological Concerns with the MI Estimator**: The validity of the paper's core metric, Mutual Information, is potentially compromised by the methodology used for its estimation. The authors state in Section 3.2 that due to the miscalibration of smaller models, they used a larger 7B "proxy model" to evaluate the log probabilities for all compressors. This introduces a significant confounding variable: it is unclear whether the MI metric is measuring the true information retained by the compressor or simply the compressor's ability to generate text that is probable under a different, larger model. This methodological choice could affect the integrity of all subsequent conclusions that rely on the MI metric.
* **Oversimplification of Agentic Workflows**: The study primarily focuses on a simple, one-shot "compress-then-predict" workflow. While this is a common pattern, it doesn't capture the complexity of more advanced, iterative agentic systems where the predictor might re-query the compressor or where multiple agents collaborate over several turns. The conclusions drawn, particularly about the primacy of the compressor, may not generalize to these more dynamic and complex architectures.
* **Limited Scope of "Compression"**: The experiments define "compression" primarily as summarization. However, in agentic systems, compression can take many forms, such as structured data extraction (e.g., to JSON), function call generation, or argument synthesis. The paper's key finding that larger compressors become more "concise" (fewer tokens) might not hold for these other compression modalities, where a better compression could be more detailed and thus longer.
* **Clarity of Figures and Captions**: The presentation of several key figures could be improved. Specifically, some figure captions are not fully self-contained or are difficult to read, forcing the reader to hunt for details in the main text to understand the axes and curves. For example, the captions for Figure 2 and Figure 3 are dense and require careful cross-referencing with the text, which hinders clarity and makes the results harder to interpret at a glance.

**Questions:**

See the Weaknesses section.

---

> ### Author Response · Authors · 2025-11-24
> **Response to reviewer H8yM (1)**
>
> We thank the reviewer for their time and feedback. We were inspired by their suggestions and have added additional results and experiments in the following.
>
> ### **Reliance on proxy models:**
>
> _Comment: The authors state in Section 3.2 that due to the miscalibration of smaller models, they used a larger 7B "proxy model" to evaluate the log probabilities for all compressors. This introduces a significant confounding variable …_
>
> **Response:** We agree and provide additional experiments that show:
> 1. When sweeping across proxy models of different families, we find same scaling between model size and mutual information. The choice of proxy model only affects the vertical offset.
> 2. The proxy models are in fact only needed at the 1B scale. When you go beyond 1B, you can use the actual model’s log-probabilities, with no proxies whatsoever. We hypothesize that the next generation of 1B models will be calibrated enough so that proxy models will not be needed.
>
> Please see Global Response and response to reviewer “xvWko”.
>
> ### **Oversimplification of Agentic Workflows:**
>
> _Comment: The study primarily focuses on a simple, one-shot "compress-then-predict" workflow. While this is a common pattern, it doesn't capture the complexity of more advanced, iterative agentic systems where the predictor might re-query the compressor or where multiple agents collaborate over several turns. …_
>
> **Response:** We agree with the reviewer. Indeed, multi-turn workflows are ubiquitous and we ran a new experiment to provide an initial understanding of how mutual information scales with multi-turn agent interactions. We see that going beyond a single turn increases mutual information. Going beyond two turns is less clearly helpful. (see Global Response and response to reviewer “nAdb”).
>
> ### **Limited Scope of "Compression”:**
>
> _Comment: The experiments define "compression" primarily as summarization. However, in agentic systems, compression can take many forms, such as structured data extraction (e.g., to JSON), function call generation, or argument synthesis. …_
>
> **Response:** Thank you for pointing out these forms of compression, and we now call out these forms of compression as a promising direction for future work. For us, “compression” stands for any set of output tokens from the compressor model, whether structured or unstructured. For simplicity, we focused on the intuitive unstructured case, for which there are a variety of well established benchmarks (FinanceBench, LongHealth, QASPER).
>
> ### **Clarity of Figures and Captions:**
>
> _Comment: The presentation of several key figures could be improved. Specifically, some figure captions are not fully self-contained or are difficult to read, forcing the reader to hunt for details in the main text to understand the axes and curves. …_
>
> **Response:** As per your feedback, we have improved the clarity of ALL our figures and figure captions.

---

### Official Review · Reviewer_nAdb · 2025-10-18

**Soundness:** 2
**Presentation:** 2
**Contribution:** 2
**Rating:** 4
**Confidence:** 4

**Summary:**

Agentic language model systems have become prevalent in AI workflows, often featuring smaller compressor models that distill raw contexts into summaries for larger predictor models to generate answers. The motivation stems from the ad-hoc nature of designing these systems, lacking principled ways to evaluate compressor quality independently or attribute performance gains. Challenges include the difficulty in quantifying compression efficacy without task-specific metrics and the computational intractability of exact mutual information calculations. The paper proposes an information-theoretic framework, introducing a Monte Carlo estimator for mutual information between context and compression, along with rate-distortion analysis to predict downstream performance, validated through empirical studies on four datasets showing that scaling compressors yields greater efficiency and accuracy than scaling predictors.

**Strengths:**

1. Empirical results demonstrate clear scaling laws favoring larger compressors. Larger models produce more concise yet informative summaries, leading to sublinear compute cost increases. These findings offer practical guidance for optimizing agentic systems in resource-constrained environments.

2. The rate-distortion analysis reveals strong correlations between information rate and accuracy. Bit efficiency metrics predict performance with high fidelity, as shown by R-squared values up to 0.71. This tool allows for quick assessment of compressor-predictor pairings.

3. Experiments across multiple model families highlight compressor choice as the dominant factor. Qwen-2.5 models show superior efficiency compared to Llama and Gemma-3. Such comparisons inform model selection for specific deployment scenarios.

**Weaknesses:**

1. The mutual information estimator relies on proxy models for smaller LMs. This introduces potential biases in log-probability evaluations. Such approximations may not fully capture the true information content.

2. Analysis is limited to non-reasoning GPT-style models. This restricts generalizability to reasoning-augmented or multi-turn agentic systems. Future work is needed to extend to more complex architectures.

3. The framework assumes single-round communication. This overlooks iterative multi-agent workflows common in practice. Extending to multi-round scenarios could reveal different scaling behaviors.

4. The claimed communication efficiency by text compressor actually is little, because the texts occupy much smaller memory than models or features. Considering the extra computational costs, the communication time saved by the compressor is not comparable to the extra computation time.

5. Computational estimates use simplified FLOPs calculations. These may not account for real-world hardware variations or quantization effects. More precise benchmarking on actual devices is necessary.

**Questions:**

See weaknesses.

---

> ### Author Response · Authors · 2025-11-24
> **Response to reviewer nAdb (1)**
>
> We appreciate the reviewer’s careful review and feedback of our work. Please find our responses below.
>
> ### **Reliance on proxy models:**
>
> _Comment: The mutual information estimator relies on proxy models for smaller LMs. This introduces potential biases in log-probability evaluations. …_
>
> **Response:** We agree and therefore provide additional experiments to understand the effect of proxy model choice on mutual information estimates. However, since proxy models are necessary for mutual information estimation in smaller LMs, we run an ablation over three proxy LMs from different models families and find that while different proxies introduce a fixed offset in MI cures, the scaling trends and slope remain unaffected by proxy model choice. However, we agree that the reliance on proxy models is not ideal. Therefore, we repeat our compressor scaling experiments on larger, better-calibrated compressors and observe consistent scaling trends (see Global Response and response to reviewer “xvWko”).
>
> ### **Limited to non-reasoning GPT-style models:**
>
> _Comment: Analysis is limited to non-reasoning GPT-style models. This restricts generalizability to reasoning-augmented or multi-turn agentic systems. …_
>
> **Response:** This is an important suggestion that led to a set of new reasoning and mixture-of-experts compressor model experiments. Larger reasoning compressors achieve higher QA accuracy and higher mutual information. However, they do not necessarily produce more concise compressions. Please see the Global Response for more details.
>
> ### **Single-round communication overlooks iterative workflows:**
>
> _Comment: The framework assumes single-round communication. This overlooks iterative multi-agent workflows common in practice. …_
>
> **Response:** Thank you for highlighting this important class of workflows! While the focus of this work is on the compression-prediction workflow, which addresses static, single-round tasks making up ~80% of LLM tasks today [1], we agree that understanding multi-turn workflows is a really interesting problem. Your feedback led to a new experiment to provide an initial understanding of how mutual information scales with multi-turn agent interactions. We find that adding a second round of communication increases mutual information carried by the compressors, but after two rounds there is no improvement (see Global Response).
>
> [1] https://arxiv.org/abs/2508.15361
>
> ### **Improved communication efficiency is negligible:**
>
> _Comment: The claimed communication efficiency by text compressor actually is little, because the texts occupy much smaller memory than models or features. …_
>
> **Response:** We thank the reviewer for highlighting this trade-off, and have removed all claims of communication efficiency from the main text. As the reviewer points out, it is not the number of tokens that is significant, but rather the FLOPs and memory footprint of the underlying LMs. These metrics scale with the number of output tokens: longer generations means that larger KV caches have to be computed, with slower decode and higher costs for frontier models.
>
> ### **FLOPs are simplified estimates for computation:**
>
> _Comment: Computational estimates use simplified FLOPs calculations. These may not account for real-world hardware variations or quantization effects. …_
>
> **Response:** This is an important consideration that is highly relevant for real-world deployments.
> For compressor models that we deploy ourselves on our own hardware, we report FLOPs as a rough proxy for computational performance. While quantization can alter FLOPs, it also affects model performance that is not within the scope of this paper.

---

### Official Review · Reviewer_vWko · 2025-10-18

**Soundness:** 2
**Presentation:** 2
**Contribution:** 2
**Rating:** 4
**Confidence:** 3

**Summary:**

The paper explores agentic language model systems where smaller compressor models distill raw contexts into compact summaries for larger predictor models to generate user responses. Challenges include the difficulty in measuring compression efficacy without full system evaluation and understanding scaling impacts on compute efficiency. Solutions involve introducing a mutual information estimator for task-agnostic compression assessment, conducting rate-distortion analysis to link information retention to downstream performance, and empirical studies across datasets showing benefits of scaling compressors over predictors.

**Strengths:**

1. The mutual information estimator offers a practical tool for evaluating compression without requiring downstream tasks. It computes effectively using modern inference servers. This method provides insights comparable to perplexity for predictors.

2. Rate-distortion analysis establishes strong correlations between information rate and task performance. It serves as a reliable proxy for system efficacy. The framework guides optimization of communication in agentic designs.

3. Empirical findings reveal a clear hierarchy where compressor family outweighs predictor size in importance. Scaling compressors yields greater gains than predictors. This principle supports front-loading compute into local devices.

4. Application to Deep Research pipelines achieves near-frontier accuracy with small local compressors. It reduces API costs significantly while maintaining quality. The approach demonstrates real-world utility in multi-agent workflows.

**Weaknesses:**

1. Reliance on proxy models for mutual information estimation in smaller LMs introduces potential biases. The approximation may not fully capture information dynamics. This could compromise the accuracy of task-agnostic evaluations.

2. Restriction to non-reasoning GPT-style models limits the study's scope. Reasoning tokens require separate analysis not covered here. The postponement leaves gaps in applying findings to advanced agentic architectures.

3. Use of subsampled or synthetic datasets may not reflect real-world variability. For instance, WildChat employs grouped synthetic users. This setup could skew scaling observations toward artificial patterns.

4. Assumption of Gaussian form in rate-distortion fitting oversimplifies LM data distributions. Distortion measured as 1-accuracy ignores nuances in evaluation. More flexible models might yield deeper insights into trade-offs.

5. Cost estimates depend on specific API rates and hardware assumptions that evolve rapidly. They overlook quantization impacts on deployment. This diminishes the practical longevity of economic recommendations.

**Questions:**

No other questions.

---

> ### Author Response · Authors · 2025-11-24
> **Response to reviewer vWko (1)**
>
> We thank the reviewer for their thoughtful comments and are glad that they found the problem interesting and practical.  We address the reviewer’s concerns below.
>
> ### **Reliance on proxy models:**
>
> _Comment: Reliance on proxy models for mutual information estimation in smaller LMs introduces potential biases…_
>
> **Response:** We agree with the reviewer and therefore provide additional experiments to understand the effect of proxy model choice on mutual information estimates. Our experiments reveal that the proxy models are only needed for small, uncalibrated LMs at the 1B scale. For models beyond 3B, there is no need for proxy models at all, as we show with the Qwen-3 model family. To bolster our existing findings, we sweep over three proxy LMs from different model families and find that while different proxies introduce a fixed offset in MI cures, the scaling trends and slope remain unaffected by proxy model choice (see Global Response).
>
> ### **Restriction to non-reasoning GPT-style models:**
>
> _Comment: Restriction to non-reasoning GPT-style models limits the study's scope. Reasoning tokens require separate analysis not covered here. …_
>
> **Response:** This is an important suggestion that led to a set of new reasoning and mixture-of-experts compressor model experiments. Larger reasoning compressors achieve higher QA accuracy and higher mutual information. However, they do not necessarily produce more concise compressions. Please see the Global Response for more details.
>
> ### **Use of subsampled or synthetic datasets:**
>
> _Comment: Use of subsampled or synthetic datasets may not reflect real-world variability. For instance, WildChat employs grouped synthetic users._
>
> **Response:** First, we emphasize that our results hold across both synthetic (WildChat, FineWeb) and real-world datasets (LongHealth, FinanceBench). In addition to our existing datasets, we now replicate our compressor scaling results on a third non-synthetic dataset, QASPER, that involves summarizing scientific research papers. (see Global Response.)
>
> ### **Assumption of Gaussian form oversimplifies LM data distributions:**
>
> _Comment: Assumption of Gaussian form in rate-distortion fitting oversimplifies LM data distributions. Distortion measured as 1-accuracy ignores nuances in evaluation. More flexible models might yield deeper insights into trade-offs._
>
> **Response:** We acknowledge this limitation in the main text (Section 2.2) to motivate deeper analysis in future works.
>
> ### **Cost estimates depend on API rates and hardware:**
>
> _Comment: Cost estimates depend on specific API rates and hardware assumptions that evolve rapidly. They overlook quantization impacts on deployment. …_
>
> **Response:** We agree with the reviewer that compute and API dollar costs evolve quickly. We now emphasize in the main text that API pricing as a metric for compute cost in the DeepResearch setting is a snapshot of relative model costs in time. Note however, that we use the dollar cost metric only for frontier models. Since we neither know the exact model sizes nor the extent of quantization used by the frontier model providers, we take dollar cost as a proxy for FLOPs (their cost of serving it). While absolute prices do evolve rapidly, the relative costs within the same generation tend to remain stable and carry over to newer model generations, making the comparison meaningful for understanding broader design and architectural trade-offs.

---

### Official Review · Reviewer_DhNP · 2025-10-25

**Soundness:** 3
**Presentation:** 3
**Contribution:** 3
**Rating:** 6
**Confidence:** 2

**Summary:**

This paper examines “compressor–predictor” architectures for agentic LLM systems (e.g., Deep Research pipelines), where a smaller LM compresses the raw context and a larger LM performs reasoning/generation.
The authors bring an information-theoretic lens, framing the compressor as a noisy channel and using mutual information (MI) as a task-agnostic measure of compression quality.

**Strengths:**

Applying MI and rate–distortion theory to compressor–predictor pipelines is interesting and provides a principled view on design trade-offs.

The MI estimator is implementable on real inference stacks; could directly help practitioners evaluate compressors without full end-to-end sweeps.

The four distilled principles are easy to interpret and potentially impactful in industry.

**Weaknesses:**

- The information-theoretic analysis stops at empirical correlation and heuristic exponential fitting. No formal connection is established between MI and downstream accuracy beyond observed correlation
- No ablation quantifying the error or variance of the MI estimator.
- Experiments focus on one-shot, GPT-style instruction models; no evidence for robustness to multi-turn reasoning agents, tool-using agents, or MoE architectures.
- Some datasets or QA pairs are synthetic (GPT-generated), which may bias outcomes toward LMs similar to the generator.
- Downstream metrics limited to accuracy/perplexity; no human evaluation or diverse NLG metrics (e.g., ROUGE, BLEU, factuality) for generative tasks.
- Failure mode analysis is qualitative only; lacks quantitative measurement of how much each error type contributes to performance loss.

**Questions:**

How sensitive is the MI–accuracy correlation to the choice of proxy model for log-prob estimation?
Can the MI estimator be extended to handle reasoning chains or intermediate tool calls where information is not localized in a contiguous context?
Did you attempt any adjustments to the distortion metric beyond accuracy (e.g., weighted accuracy for partial credit or semantic similarity for generative tasks)?
For Deep Research, what happens if the predictor is also small and local — does the compressor's advantage still hold?

---

> ### Author Response · Authors · 2025-11-24
> **Response to reviewer DhNP (1)**
>
> We thank the reviewer for their invaluable feedback and suggestions, which have inspired several additional experiments that we outline in the following.
>
> ### **No formal connection between MI and downstream accuracy:**
>
> _Comment: No formal connection is established between MI and downstream accuracy beyond observed correlation_
>
> **Response:** You point out a valuable gap in our current information-theoretic analysis. To establish a more formal framing of our mutual information estimator and information-theoretic framework, we conduct additional formal analyses in B.2, B.3, B.4, which derive lower bounds on I(X; Y) through the error rate/observed accuracy. We also characterize the bounds of mutual information and our estimator. Additionally, B.5 provides analysis of the rate-distortion model under an idealized Gaussian assumption that connects accuracy and mutual information through the rate-distortion model.
>
> ### **No ablation quantifying variance of the MI estimator:**
>
> _Comment: No ablation quantifying the error or variance of the MI estimator._
>
> **Response:** We thank the reviewer for highlighting this gap in the presentation of our MI estimator.
> To address this, we run a controlled synthetic experiment to quantify the error and variance of our estimator given the true mutual information in closed form. We construct a high-dimensional linear-Gaussian model, and sweep over dimensionality d and number of samples N and M. We find that our estimator shows a negative bias that grows with dimension (due to being upper bound by the number of samples log(N)) consistent with previous research [2]. Our estimator exhibits relatively stable variance (given that we are not upper bound by log(N)).
>
> [2] https://arxiv.org/pdf/1910.06222
>
> ### **Experts focus on one-shot, GPT-style instruction models:**
>
> _Comment: Experiments focus on one-shot, GPT-style instruction models; no evidence for robustness to multi-turn reasoning agents, tool-using agents, or MoE architectures._
>
> **Response:** We agree with you and thus added a set of new experiments with Qwen-3 (reasoning and mixture-of-experts compressor models), and multi-turn experiments. Larger reasoning compressors achieve higher QA accuracy and higher mutual information. However, they do not necessarily produce more concise compressions. Please see the Global Response for more details.
>
> ### **Use of synthetic datasets:**
>
> _Comment: Some datasets or QA pairs are synthetic (GPT-generated), which may bias outcomes toward LMs similar to the generator._
>
> **Response:** Thank you for highlighting the limitations of synthetic datasets!
> While 1-2 of our datasets are synthetic, our experiments also include non-synthetic benchmarks such as LongHealth and FinanceBench. To further strengthen our claims, we additionally reproduce our scaling experiments on a third non-synthetic dataset QASPER, which is a common real-world problem setting (summarizing scientific research papers). We find that our  scaling results and design principles are robust across both synthetic and non-synthetic benchmarks (see Global Response and response to reviewer “xvWko”).
>
> ### **Downstream metrics limited to accuracy/perplexity:**
>
> _Comment: Downstream metrics limited to accuracy/perplexity; no human evaluation or diverse NLG metrics (e.g., ROUGE, BLEU, factuality) for generative tasks._
>
> **Response:** Your feedback motivated a new distortion measurement based on semantic similarity (embedding ground truth answers and predictions, and computing cosine similarity). The resulting plots reproduce the characteristic rate-distortion curves, which we add to our rate-distortion analysis (E.2.2). We aim to capture more nuanced differences in outputs (see Global Response).
>
> ### **Failure mode analysis is qualitative only:**
>
> _Comment: Failure mode analysis is qualitative only; lacks quantitative measurement of how much each error type contributes to performance loss._
>
> **Response:** The reviewer highlights an interesting point of analysis and valuable direction for future work. We remain qualitative in our failure mode analysis to clarify the types of error that commonly emerge in compression-prediction workflows. Measuring how much each error type contributes to performance loss would be nontrivial and beyond the scope of this work.

---

> ### Author Response · Authors · 2025-11-24
> **Response to reviewer DhNP (2)**
>
> ## Questions:
>
> #### _How sensitive is the MI–accuracy correlation to the choice of proxy model for log-prob estimation?_
>
> According to our new experiments, not at all. The choice of proxy model introduces a fixed vertical offset in the MI curves that is constant across estimates, but does not impact scaling rates.
>
> #### _Can the MI estimator be extended to handle reasoning chains or intermediate tool calls where information is not localized in a contiguous context?_
>
> Yes, there are interesting ways to synthesize reasoning chains and tool calls into text outputs that can be directly provided to the MI estimator. More specifically, a tool call is a way to interface with the data plane (a web search), and is perfectly compatible with our framework. For simplicity, we do not explicitly explore that as a mode of compression.
>
> #### _Did you attempt any adjustments to the distortion metric beyond accuracy (e.g., weighted accuracy for partial credit or semantic similarity for generative tasks)?_
>
> Yes, we added a semantic similarity metric using cosine similarity between prediction and ground-truth answer embeddings. See Global Response.
>
> #### _For Deep Research, what happens if the predictor is also small and local — does the compressor's advantage still hold?_
>
> We investigate this setting (Llama-3.1 8B predictor) in our rate-distortion analysis (Figure 3, Figure 6). We find that increasing compressor size yields less significant improvements in downstream QA accuracy. As a consequence, it is less FLOPs-efficient to increase compressor size since we are limited by predictor capacity.

---

### Author Response · Authors · 2025-11-24
**Global Reponse**

We thank the four reviewers for the time and effort reviewing our work and providing invaluable feedback. We’ve incorporated the reviewers’ feedback in our revision and include new experiments.

We appreciate the positive feedback from all the reviewers:
- The paper makes an “interesting” [DhNP] and “significant conceptual contribution” [H8yM] to a “very important problem” [H8yM].
- The mutual information estimator is “practical” [vWko], “implementable” [DhNP], and offers a “novel and principled framework” [H8yM] in a space that is based on “ad-hoc trial-and-error” [H8yM]. It provides a “reliable proxy for system efficacy” [vWko] that is “comparable to perplexity” [vWko].
- The authors derive “clear, actionable, and somewhat counter-intuitive design principles” [H8yM] that have “significant practical implications for system design” [H8yM] and are “potentially impactful in industry” [DhNP].
- The results—achieving “near-frontier accuracy with small local compressors” [vWko]—are “thorough” [H8yM], supported by  “strong” [vWko, nAdb, H8yM] evidence“ and an “extensive” [H8yM] set of experiments.

The reviewers also pointed to a few areas of potential improvement. We highlight the major improvements we have made to our experiments in response to their feedback. We also provide further improvements in the reviewer-specific responses.

**Robustness towards the choice of proxy model [vWko,nAdb,H8yM].**
All reviewers raised the usage of proxy models as a concern, which propelled us to run a new set of experiments. We provide analyses evaluating whether our mutual information estimator is robust towards the proxy model used to compute log probabilities. First, we analyze the effect of three different proxy model choices in E.1.4. Across all three proxy models, we observe the expected compressor scaling behavior: larger compressors yield higher mutual information. The proxy model choice introduces a fixed vertical offset in the MI curves that is consistent across estimates, but does not impact scaling rates.

Furthermore, we also evaluate the mutual information estimates for compressors that are sufficiently large and well-calibrated (Qwen-3 4B, 8B, 30B-A3B) using their internal log probabilities. We find that scaling trends are consistent with our previous observations, and that proxy models are only necessary for smaller LMs that are often miscalibrated (assign high likelihoods to nonsensical token sequences).

**Adding a new analysis of reasoning models [vWko,nAdb,DhNP].**
In our revision, we expand our compressor scaling analysis to a new model generation Qwen-3, that includes both reasoning and mixture-of-experts (MoE) models, allowing us to quantify their effects. We separate the reasoning tokens from answer tokens and measure QA accuracy on LongHealth, compression length, and estimate mutual information. We observe that larger reasoning compressors generate compressions that yield higher downstream accuracy and carry more mutual information with the input. However, they do not necessarily produce more concise compressions, unlike the rest of our results, which highlights an interesting direction for future research.

In an experiment in E.1.5, we compared MoE to a dense model with the same number of active parameters. When comparing the number of active parameters, MoEs produce more concise outputs that carry significantly more mutual information, indicating an open direction for future work.

**Multi-turn workflows [nAdb,DhNP,H8yM].**
Inspired by the reviewers, we evaluate multi-turn compression-prediction workflows on LongHealth. We give a Llama-3.1 405B predictor the ability to query a Llama-3.2 3B compressor for additional information three times. At each turn, the predictor synthesizes the previous compression with the new queried information, and issues a follow-up query for additional information. In the multi-turn setting, mutual information increases when going beyond a single turn (but no increase from two to three turns). This opens wonderful grounds for followup in future research.

**Alternative measurements of distortion [DhNP].**
Inspired by the reviewers’ suggestions, we provide additional results showing that the trends hold with an alternative, more general, definition of distortion. We embed both prediction and target answer and measure the cosine similarity between the embeddings. We observe characteristic rate-distortion curves with the same ordering of predictor model sizes (E.2.2).

**Non-synthetic datasets [vWko,DhNP].**
In response to the reviewers’ concerns, we have expanded our compressor scaling analysis to a third non-synthetic long-context QA dataset—QASPER. We repeat our full compressor scaling analysis on non-reasoning Qwen-2.5 and Llama-3, and reasoning Qwen-3 compressor models. We observe scaling trends consistent with our previous findings: larger compressors achieve higher QA accuracy and produce more concise compressions that carry more mutual information (E.1.1).

---

### Comment · Area_Chair_4Ppk · 2025-11-26
**Reviewer & Author Discussion**

Dear Reviewers,

We kindly encourage you to review and respond to the authors’ rebuttals. Your timely feedback is important for ensuring a fair and thorough review process. Thank you for your contributions to ICLR 2026.

Thank you very much for your time and support.

Best regards,

 Area Chair 4Ppk

---

### Author Response · Authors · 2025-12-04
**Global Response to Reviewers and Area Chair**

We thank the AC for their message and thank all reviewers for the time and care dedicated to our submission. We carefully addressed all points raised by the reviewers with new analyses, experiments, and in-text clarifications. We have uploaded a revised PDF with additional explanations and details in the main body, and further experiments and results added to the appendix.

Key improvements include:
- **Proxy model robustness**: We validate that proxy model choice only yields a vertical offset consistent across all mutual information estimates. The observed scaling behavior is robust to proxy choice. Furthermore, we confirm that no proxy models are needed for well-calibrated compressors.
- **New analysis of reasoning models**: We further expand our compressor scaling analysis to include both reasoning and Mixture-of-Experts compressors from the Qwen-3 model generation. Our results reveal new trade-offs in conciseness, accuracy, and mutual information.
- **Multi-turn workflows**: We introduce a new multi-turn compression-prediction experiment mirroring more complex agentic systems. We find that mutual information increases with additional turns (with diminishing returns beyond two).
- **Additional non-synthetic dataset**: We expand our compressor scaling analysis to a new third non-synthetic dataset QASPER. We observe compressor scaling behavior consistent with our previous findings.
- **Additional measurements of distortion**: We add a measurement of distortion based on semantic similarity.
- **Additional formal analyses**: We also quantify the variance and bias of the MI estimator using synthetic controlled experiments, and expand the formal connection between MI and accuracy.

We hope that these revisions address the reviewers’ feedback and complement the rigor and contribution of our work.

---

### Meta-Review · Area_Chair_8Ux1 · 2026-01-07

**Summary:**

The reviewers generally recognized the paper's novelty in applying information theory to agentic system design and the practical value of the "compressor-predictor" scaling laws. However, they raised significant concerns regarding the methodology, specifically the reliance on a "proxy model" for estimating Mutual Information (MI) in small compressors, which could introduce bias. Reviewers also questioned the generalizability of the findings, noting the focus on non-reasoning, single-turn GPT-style models and the use of synthetic datasets. Additional concerns included the lack of formal theoretical grounding for the MI-accuracy connection and the validity of communication efficiency claims.

**Reviewer Concerns:**

- Addressed by the rebuttal:
  - Proxy Model Reliability: The authors demonstrated that using different proxy models only introduces a vertical offset in MI curves without affecting scaling trends, and that proxies are unnecessary for well-calibrated models like Qwen-3.
  - Model Generalizability: The authors expanded experiments to include reasoning models and Mixture-of-Experts (MoE) from the Qwen-3 family, showing that larger reasoning compressors also improve downstream accuracy.
  - Multi-turn Workflows: A new experiment showed that MI increases with a second turn of interaction, addressing the concern about single-shot limitations.
  - Synthetic Data: The authors validated their scaling laws on QASPER, a third non-synthetic dataset, reinforcing robustness.
  - Formal Analysis: New theoretical work quantified the variance/bias of the estimator and derived lower bounds connecting MI to accuracy.
  - Communication Efficiency: The authors conceded that text compression does not significantly reduce communication latency compared to compute, removing this claim.
- Still outstanding:
  - Complex Dynamic Workflows: While the multi-turn experiment provides a "first step," the full complexity of iterative, multi-agent collaboration (beyond simple query-response) remains an area for future work rather than being fully solved.
  - Cost Metrics: The reliance on specific API pricing snapshots for cost-efficiency claims remains a limitation inherent to the field.

**Reviewer Scores:**

- Reviewer DhNP (Score: 6): Likely to increase to 7. Their requests for formal analysis, variance quantification, and diverse metrics were explicitly and thoroughly addressed.

- Reviewer vWko (Score: 4): Likely to increase to 6. The addition of reasoning models and analysis of proxy model robustness directly resolved their primary grounds for the lower score.

- Reviewer nAdb (Score: 4): Likely to increase to 6. The inclusion of multi-turn experiments and reasoning models, along with the correction on communication efficiency, addressed their key weaknesses.

- Reviewer H8yM (Score: 6): Likely to increase to 7. They viewed the paper as a "significant conceptual contribution" and their main methodological concern about proxy models was effectively rebutted.

---

### Decision · Program_Chairs · 2026-01-26

Accept (Poster)